# Global nighttime atomic oxygen abundances from GOMOS hydroxyl airglow measurements in the mesopause region

Qiuyu Chen[1, 2], Martin Kaufmann[1, 2], Yajun Zhu[1, 3], Jilin Liu[1, 2], Ralf Koppmann[2], and Martin Riese[1, 2]

[1]Institute for Energy and Climate Research, Forschungszentrum Jülich, Jülich, Germany
[2]Institute for Atmospheric and Environmental Research, University of Wuppertal, Wuppertal, Germany
[3]State Key Laboratory of Space Weather, National Space Science Center, Chinese Academy of Sciences, Beijing, China

**Correspondence:** Yajun Zhu (y.zhu@swl.ac.cn)

**Abstract.** This paper presents a new dataset of nighttime atomic oxygen density [O], derived from OH(8–4) ro-vibrational band emissions, using a non-local thermal equilibrium model, with the aim of offering new insight into the atomic oxygen abundances in the mesopause region. The dataset is derived from the level-1 atmospheric background measurements observed by the GOMOS instrument on board Envisat, with the SABER measurements for the atmospheric background. Raw data are reprocessed into monthly zonal mean values in 10° latitude bins with a fixed altitude grid of 3 km. The dataset spans from 70°S to 70°N in latitude and from 80 km to 100 km in altitude, covering a time period from May 2002 to December 2011 at local times from 10 p.m. to 12 p.m..

The atomic oxygen density peaks at about 95 km and the highest values are in the range of $3$–$8\times10^{11}$ atoms cm$^{-3}$, depending on latitude and season. There is a rapid decrease of [O] below the peak. The annual oscillation (AO), semiannual oscillation (SAO), and the solar cycle impact are distinguished from the [O] longtime series variations. This new GOMOS [O] dataset conforms to other published datasets and is consistent with the [O] datasets obtained from the SCIAMACHY OH airglow measurements to within about $\pm20\%$.

## 1 Introduction

In the middle and upper atmosphere, atomic oxygen (O) is mainly produced by the photolysis of molecular oxygen and of ozone, and transported downward by diffusion and mixing from the thermosphere to the mesopause. Its lifetime varies from over one week at 100 km to around one day at 80 km due to its increasing chemical loss rate with decreasing altitude (Brasseur and Solomon, 2005). Atomic oxygen is one of the most abundant reactive trace species in the upper mesosphere/lower thermosphere (MLT) region and plays a crucial role in the photochemical equilibrium and energy balance of this region. Most exothermic chemical reactions, which heat the MLT region, are associated with atomic oxygen (Brasseur and Offermann, 1986; Riese et al., 1994; Mlynczak et al., 2013c). The collisions between O and infrared-active greenhouse gases like $CO_2$ also predominantly lead to radiative cooling in this region (Mlynczak et al., 2013a).

The measurement of atomic oxygen dates back to before the satellite era when the MLT region was explored by means of sounding rocket experiments, hosting resonance fluorescence instruments, or mass spectrometers (e.g. Dickinson et al., 1974, 1980; Sharp, 1980; Offermann et al., 1981; Sharp, 1991). They are capable of providing direct in situ measurements of atomic

oxygen, although it is difficult to obtain a consistent global picture of absolute density values from these measurements, which differ by a factor of more than 40 (Offermann et al., 1981; Sharp, 1991).

However, these measurements lead to the development of photochemical models of the Earth′s day- and nightglow, which enables the use of proxies of the atomic oxygen abundance obtained from satellite observations. Suitable proxies are airglow emissions (e.g. $OH^*$, $O_2^*$, $O(^1S)$) and thermal emissions (e.g. $O_3$ at 9.6 $\mu$m), in combination with corresponding photochemical models. The hydroxyl (OH) airglow emissions are associated with the spontaneous radiative transitions of excited $OH^*$ radicals. These $OH^*$ radicals are mainly produced by the chemical reaction of ozone with atomic hydrogen. Highly excited molecular oxygen $O_2^*$ in a metastable state is generated from atomic oxygen recombination and can be de-excited by O or $O_2$, while the $O(^1S)$ and $O_2$ A-band emissions are radiated from the products. These airglow emissions rely on the atomic oxygen recombination or ozone destruction, and can be recognized as a kind of chemical afterglow. Therefore, they are frequently used as a proxy to retrieve atomic oxygen.

More recent measurements were conducted by the Sounding of the Atmosphere using Broadband Emission Radiometry (SABER) instrument on the Thermosphere-Ionosphere-Mesosphere Energetics and Dynamics (TIMED) satellite. The instrument detects $OH^*$ nightglow radiances at 2.0 and 1.6 $\mu$m as well as $O_3$ thermal emissions at 9.6 $\mu$m (Smith et al., 2010; Mlynczak et al., 2013b, 2018; Panka et al., 2018). The Scanning Imaging Absorption Spectrometer for Atmospheric CHartographY (SCIAMACHY) instrument on the European Environmental Satellite (Envisat) measures the $O(^1S)$ green line at 557.7 nm and a broad range of $OH^*$ airglow emissions (Kaufmann et al., 2014; Lednyts'kyy et al., 2015; Zhu et al., 2015; Zhu and Kaufmann, 2018). The Optical Spectrograph and Infrared Imager System (OSIRIS) instrument on the Odin satellite probes the $O_2$ A-band at 762 nm and $OH^*$ airglow at 725–745 nm and 770–815 nm (Sheese et al., 2011, 2014). During the period 1991–1995, the Wind Imaging Interferometer (WINDII) instrument on board the Upper Atmosphere Research Satellite (UARS) also observed the $O(^1S)$ green line and OH(8–3) band emissions at 734 nm (Russell and Lowe, 2003; Russell et al., 2005). Other instruments include the High Resolution Doppler Imager (HRDI) on board UARS, which also observes $O_2$ A-band emissions (Hays et al., 1993); the Imager of Sprites and Upper Atmospheric Lightning (ISUAL) instrument on board the FORMOSAT-2 satellite, which detects the $O(^1S)$ green line emissions (Gao et al., 2012); and the Solar Mesosphere Explorer (SME) spacecraft, which measures the OH(7–5) band emission at 1.87 $\mu$m (Thomas, 1990).

While various datasets are consistent in terms of the overall profile shape of derived [O] densities, some discrepancies still exist (Mlynczak et al., 2013c, a, b; Kaufmann et al., 2014; Mlynczak et al., 2018; Zhu and Kaufmann, 2018; Panka et al., 2018). The radiometric calibration of the instruments or differences in airglow model parameters are potential reasons. Therefore, no common consensus has generally been reached with regard to these aspects. Some new findings on airglow relaxation modeling and reaction kinetic parameters were recently published. A new pathway, in which highly vibrationally excited OH radicals (v≥5) are deactivated by atomic oxygen to a lower state ($0 \leq v' \leq v-5$) is proposed and discussed (Sharma et al., 2015; Kalogerakis et al., 2016; Panka et al., 2017, 2018; Fytterer et al., 2019; Kalogerakis, 2019). These results complicate the topic further.

To contribute another piece of information to the currently ongoing discussions, a new dataset derived from the OH nightglow observed by the GOMOS (Global Ozone Monitoring by Occultation of Stars) instrument on the ESA′s (European Space

Agency) environmental satellite (Envisat) during the years 2002 to 2012 is presented and discussed here. This dataset is particularly valuable in that it was obtained at the same time as the already published SABER and SCIAMACHY data, but from a different instrument with its own radiometric calibration. Emissions from OH(8-4) are used to obtain atomic oxygen abundances, which is a similar proxy as provided by the SABER and SCIAMACHY OH measurements.

5    This paper is structured as follows: the second section provides a brief introduction to the instrument and data processing procedure, followed by a section describing the airglow modeling. The derived results are shown in the fourth section, including error analysis as well as latitudinal and temporal analysis. The next section investigates the validation of the dataset in a broad context, including comparisons with the SCIAMACHY dataset and other data sources, while the final section concludes the topic with an outlook on future expectations.

## 2    Measurements and Data Preparation

### 2.1    GOMOS on Envisat

The GOMOS spectrometer is one of nine instruments on board Envisat. It is designed to monitor ozone profiles and other trace species using stellar occultation and atmospheric transmission measurements in limb-viewing mode (ESA, 2010). Envisat follows a sun-synchronous orbit with an equator crossing time (descending node) of 10 p.m. (Gottwald et al., 2011). The operation period of GOMOS dates from April 2002 to April 2012. However, there was an instrument malfunction in summer 2005, resulting in a data gap of nearly three months. The GOMOS instrument delivers one vertical profile of measurements for each occultation, and the altitude coverage spans from 5 km to 150 km, with a vertical sampling rate of better than 2 km (Kyrölä et al., 2012). It has four spectral channels in the ultra-violet to near-infrared spectral range. The spectrometer B2 (SPB2), which provides the data used in this work, covers 925–955 nm with a spectral resolution of 0.13 nm at full width half maximum (FWHM) and a sampling step of 0.056 nm (Massimo Cardaci and Lannone, 2012). The GOMOS detector has three parallel bands. The central band probes the star spectra and the upper/lower bands record the atmospheric background radiation as calibration information, as indicated in Figure 1. The altitudes of tangent points observed by three bands differ roughly by 1.7 km. OH and $O_2$ A-band are regularly detected in the upper/lower bands, together with auroral emissions and the stray light scattered by particles or molecules in the atmosphere. This dataset, which is used in our analysis, is archived in the level-1b limb dataset but not directly utilized in the operational level-2 data retrieval routines. First analyses of the extracted OH and $O_2$ A-band nightglow measurements from these background datasets were reported by Bellisario et al. (2014).

### 2.2    Data Selection and Resampling

The GOMOS data were processed with the processor version 6.01–2012. The resulting level-1b limb products have already been geolocated and calibrated (Massimo Cardaci and Lannone, 2012). The signal-to-noise ratio (SNR) of single spectra is on the order of one, and the averaging of data is required for further processing of the data.

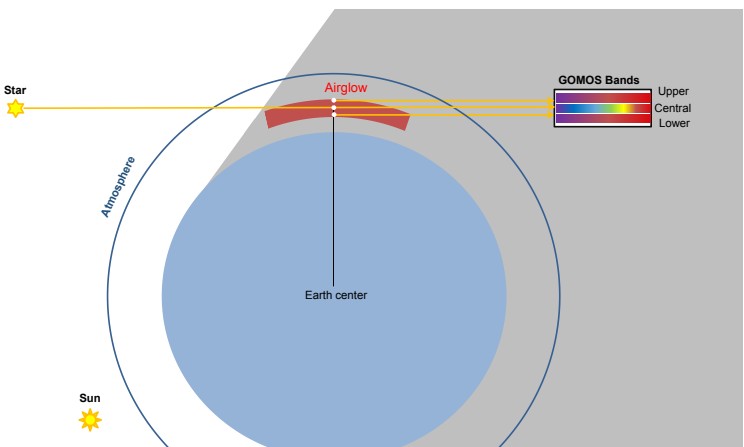

**Figure 1.** Schematic view of the GOMOS stellar occultation observations. The star transmission spectra are recorded in the central band of the instrument detector, while the atmospheric background radiation is imprinted in the upper/lower bands. The $O_2$ A-band and OH airglow emissions are detected in the limb observations. For each star spectrum, one upper and one lower spectrum are recorded simultaneously.

The raw data from level-1b limb products are first filtered with the corresponding auxiliary "quality flag" and "product confidence data"(PCD), which indicate the presence of bad pixels, saturation, cosmic rays, modulation, dark current, flat-field or vignetting correction, with only data in the normal status being kept. This is then followed by a geolocation-related selection, in which the data with ray-tracing errors are eliminated, and their star IDs and geolocation errors are restricted to within an

5 acceptable range, as recommended by Dehn (2012). The stray light entering the field of view (FOV) of the instrument affects the illumination of the spectrometer and enhances the background noise. The stray light is characterized by the illumination flags and solar zenith angle (SZA) of satellite and tangent points, which are geometrically computed. The illumination conditions of GOMOS measurements are categorized into five flags (Kyrölä et al., 2010; van Gijsel et al., 2010), and the "bright limb" flag thereof is excluded in this work. SZA > 108° is also applied as selection criteria. Near-infrared aurora at wavelengths of

10 around 939 nm and 947 nm, originating from the atomic nitrogen (N I) emissions and $N_2^+$ Meinel (2–1) band (Baker et al., 1977) are in the spectral range of SPB2. Observations in polar regions are therefore not considered in our analysis.

Due to the nature of the stellar occultation observations, the tangent points of single vertical profiles diverge significantly and are not stationary in latitude-longitude locations. In the level-2 product, they are characterized by the obliquity (Kyrölä et al., 2010), which is not available in the level-1b data. Therefore, in this work, the latitude spread of tangent points is used

instead and profiles with > 4° deviation in tangent point latitudes are disregarded to ensure that every selected profile spans a geographical area of within ±5° latitude.

The archived level-1b data are signals recorded by the detector, which must be dynamically decoded to electrons and then converted to a physical unit of flux with wavelength-specific radiometric calibration factors. The star is a point source, and part of the stellar light is spread to the lower and upper band, which is supposed to be totally imaged in the central band in an ideal

case. Considering the contamination of star leakage and residual stray light, which are assumed to be constant with altitude, the

averaged spectra from above 110 km are subtracted from each profile as background radiation. No airglow emissions are found above the region of 110 km in the GOMOS measurements. The subtraction is then followed by the individual "base" removal at each altitude layer, in which this "base" offset is the mean of residual noise of the emission lines. The processed data are resampled into monthly and zonally averaged 10° latitude bins with a fixed altitude grid of 3 km to enhance the spectra SNR and improve retrieval quality. The number of profiles selected for one sample bin (shown in Figure 2) is around 100 to 300. In order to eliminate the effect of random and systematic noise as well as outliers while retaining as many profiles as possible, the largest and smallest 1% are disregarded from the measurements at each sample bin.

Barrot et al. (2003) reported high pixel response non-uniformity (PRNU) variation of around 12% in spectrometer B (SPB). As shown in Figure 3, in the spectral range of our interest (SPB2), we found the GOMOS data shows a good agreement with the SCIAMACHY data at the spectral range of 930–935 nm, whereas the GOMOS radiances at the wavelength range of 935–955 nm are always 25–30% lower compared to the SCIAMACHY measurements, which is not understood (E. Kyrölä, personal communication, 2019). Therefore, of the entire spectral range, only the wavelength region of 930–935 nm is utilized in the retrieval to derive the atomic oxygen abundances. It includes a number of emission lines from OH(v=8–4) band, which originates from the radiative transitions of OH($v'$=8 $\longrightarrow$ $v''$=4). The dominant emission lines are mainly in the R branch with a rotational state quantum number of $K''$ = 1, 2, and 3.

The quality of the reprocessed spectra are evaluated by calculating the standard deviation (STD) of averaged spectra for each sample bin, supplemented by the SNR analysis. The calculations show that the mean STD for a typical sample bin in autumn at mid-latitudes is around $2$–$4 \times 10^9$ photons s$^{-1}$ cm$^{-2}$ nm$^{-1}$ sr$^{-1}$, and that SNR increases to more than 10 at peak altitudes and to around 3–5 at lower altitudes. A typical profile of processed hydroxyl spectra and integrated radiance are illustrated in Figure 4. Three lines are clearly visible in the spectra (left), while the emission peak layer appears at the tangent altitude of around 85 km, according to the right-hand side plot. The error bars in the right plot indicate the measurement noise for integrated radiance. The measurement noise is calculated from the standard deviation of the residual noise in the spectral range in between of the emission lines, and assumed to be the same for all wavelengths, as the intensities of remaining weak emission lines from high rotational levels in the spectral region are by several order of magnitude lower and therefore negligible. For the integrated radiance, the measurement noise is increased by a factor of $\sqrt{N}$, and N refers to the number of integrated wavelength points.

## 3  OH Airglow Modeling and Retrieval Methods

The method to derive atomic oxygen abundance relies on the chemical equilibrium between ozone production and loss during nighttime. It is also applied for the retrieval of atomic oxygen abundances from SABER (Mlynczak et al., 2018) and SCIA-MACHY (Zhu and Kaufmann, 2018) OH measurements. Ozone is produced in the three-body recombination reaction of atomic and molecular oxygen. Ozone is destroyed in reactions with atomic hydrogen and oxygen. As summarized in Table 1, most of the model parameters are adopted from Zhu and Kaufmann (2018). Additionally, the rate coefficients for the production of OH(v=8) by the collision of OH(v=9) with oxygen, and the collisional removal of OH(v=8) by atomic oxygen are obtained

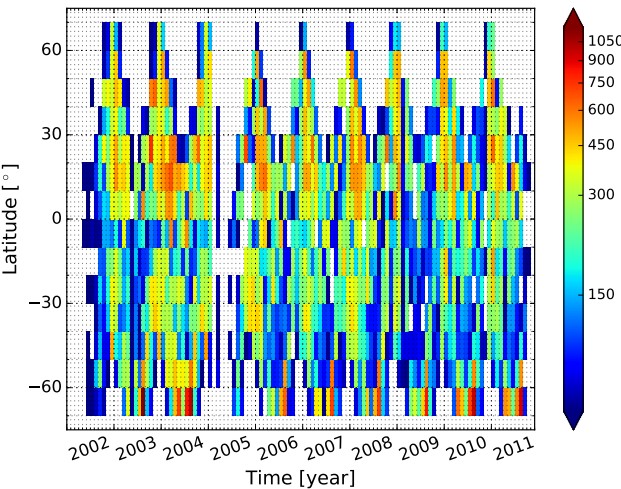

**Figure 2.** Latitudinal distribution of resampled GOMOS data available from 2002 to 2012. Colour coding indicates the number of selected profiles for each monthly and zonally averaged $10°$ latitude bin.

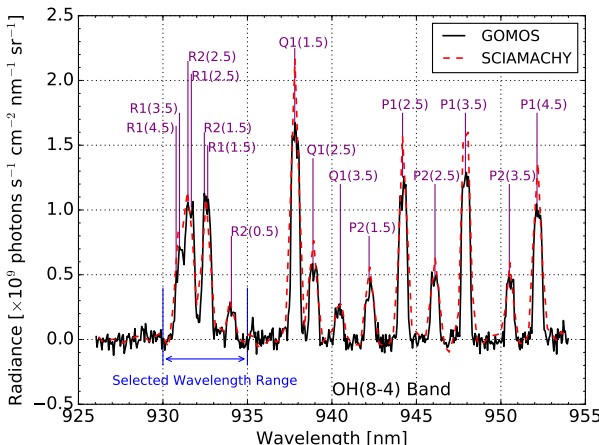

**Figure 3.** Monthly averaged spectrum from GOMOS (black solid line) for Feb. 2004 at $40°$–$50°$N and at a tangent altitude of 89.5 km. Strong emission lines from OH(8–4) band are annotated with the branch and rotational quantum numbers. The wavelength range from 930 nm to 935 nm is selected and used in the retrieval. The corresponding SCIAMACHY data (red dashed line) is also given here for comparison.

by simultaneously fitting the limb radiances of OH(9–6) and OH(8–5) bands, which are independently taken from the SCIA-MACHY measurements. These two parameters are adjusted in such a way that the ratio between the fitted radiances of the two bands is consistent with the ratio calculated from the measurements. Details about the fitting of the parameters are provided in the Appendix A.

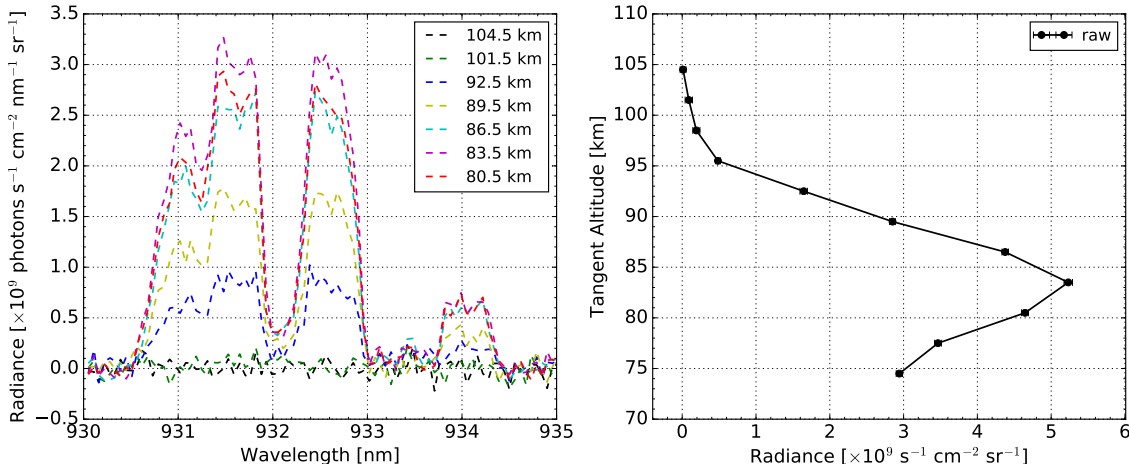

**Figure 4.** (Left) GOMOS monthly zonal mean spectra of OH(8–4) emissions at tangent altitudes as given in the figure legend for Oct. 2003 at $0°$–$10°$S and at a local time of 10–12 p.m.. (Right) The spectrally integrated radiance over 930–935 nm versus tangent altitude for the same conditions. The error bars indicate the measurement noise for integrated radiance (see text).

**Table 1.** OH airglow modeling parameters used in this study, where $k_1$ to $k_3$ represent the chemical reaction rate coefficient; $f_9$ and $f_8$ refer to the sum of Einstein coefficients of spontaneous transitions from vibrational level v=9 or 8; and $k_{N_2}$, $k_{O_2}$, and $k_O$ represent the quenching coefficients of $OH^*$ radicals by $N_2$, $O_2$, and O.

| Parameter | Process | Rate Constant | Reference |
|---|---|---|---|
| $k_1$ | $O + O_2 + M$ | $6.0 \times 10^{-34}(300/T)^{2.4} cm^6 s^{-1}$ | Sander et al. (2011) |
| $k_2$ | $H + O_3$ | $1.4 \times 10^{-10} exp(-470/T) cm^3 s^{-1}$ | Sander et al. (2011) |
| $k_3$ | $O + O_3$ | $8.0 \times 10^{-12} exp(-2060/T) cm^3 s^{-1}$ | Sander et al. (2011) |
| $f_9$ & $f_8$ | $OH^*$ nascent branching factor | 0.47 & 0.34 | Adler-Golden (1997) |
| $k_{N_2(8)}$ | $OH(8) + N_2$ | [a]$1.4 \times (7 \pm 4) \times 10^{-13} cm^3 s^{-1}$ | Adler-Golden (1997) (Table 1, measured by Dyer et al. (1997)) |
| $k_{O_2(8)}$ | $OH(8) + O_2$ | [a]$1.18 \times (8 \pm 1) \times 10^{-12} cm^3 s^{-1}$ | Adler-Golden (1997) (Table 1, measured by Dyer et al. (1997)) |
| $k_{O(8)}$ | $OH(8) + O$ | $6.5 \times 10^{-11} cm^3 s^{-1}$ | this work |
| $k_{O_2(9,8)}$ | $OH(9) + O_2 \rightarrow OH(8) + O_2$ | [a]$1.18 \times 8.9 \times 10^{-13} cm^3 s^{-1}$ | this work |
| $k_{N_2(9)}$ | $OH(9) + N_2$ | [a]$1.4 \times (7 \pm 2) \times 10^{-13} cm^3 s^{-1}$ | Kalogerakis et al. (2011) |
| $k_{O_2(9)}$ | $OH(9) + O_2$ | [a]$1.18 \times (2.2 \pm 0.6) \times 10^{-11} cm^3 s^{-1}$ | Kalogerakis et al. (2011) |
| $k_{O(9)}$ | $OH(9) + O$ | $(2.3 \pm 1) \times 10^{-10} cm^3 s^{-1}$ | Kalogerakis et al. (2016) |

[a] A low temperature scale factor, as the mesopause temperature is normally much lower than the laboratory conditions (Lacoursiére et al., 2003; Panka et al., 2017).

Atmospheric background profiles of temperature, total density, and ozone mixing ratio are taken from SABER measurements (v2.0–2016). The same latitude bins ($\pm 5°$) and local times ($\pm 1$ hour) were selected for SABER data as GOMOS data. Since

SABER cannot measure $O_2$ and $N_2$ mixing ratios, these quantities are taken from the mass spectrometer incoherent scatter (MSIS) simulation model data (Picone et al., 2002).

The inverse model applies a constrained global-fit approach following the formalism of Rodgers (2000). The Gauss-Newton iterative method in the n-form (Rodgers, 2000, p. 85) is chosen to minimize the cost function of this inverse problem. Besides, a priori information about the atmospheric state is included in the retrieval for regularization to mitigate the influence of measurement errors. The a prior information about atomic oxygen in this work is taken from MSIS model data, and the zero- and first- order Tikhonov regularization matrices (Tikhonov and Arsenin, 1977) are considered in the cost function. The a priori data about the absolute value of atomic oxygen is taken from MSIS model, which is averaged into the vertical grid of 3 km as the measurements. The first order regularization is obtained from the linear interpolation of the a priori data given on the measurement grid, i.e., no sub measurement-grid information is obtained from that data. The regularization strength depends on altitude and its main purpose is to assure meaningful values at the upper and lower boundaries of the altitude regime considered. In between, the regularization has virtually no effect on the retrieved quantities, as can be seen from the retrieval diagnostics. The vertical resolution of the retrieval results are close to the vertical grid of the measurements. The target parameters of the retrieval are the vertical profiles of atomic oxygen abundance, spectral resolution, and a wavelength shift. The latter are both altitude-independent and give a better agreement between measured and simulated spectra. The content of information in the spectra is sufficient to retrieve these additional parameters.

## 4 Results

### 4.1 Atomic Oxygen Abundances

Applying the global fitting method to GOMOS level-1b limb products, a globally distributed time series [O] dataset is derived, along with other quantities. Shown in Figure 5 (left) is a typical profile of the fitted spectra compared with the measurements. In general, simulations and measurements are in good agreement. The spectrally integrated radiances in Figure 5 (right) also show consistency. The derived oxygen densities are within an altitude range of 80 km to 100 km, covering the period from May 2002 to December 2011 and spanning local times from 10:00 p.m. to 12:00 p.m.. A typical atomic oxygen profile is shown in Figure 6 with a maximum concentration of about $3.5 \times 10^{11}$ atoms cm$^{-3}$ at 95 km. Above the maximum, there is a downward flux of atomic oxygen by diffusive transport (Swenson et al., 2018). Turbulences associated with gravity wave breaking, along with damped waves or tides, are the dynamic processes that contribute to this diffusive transport (Smith et al., 1987; Li et al., 2005). Below the maximum, there is a rapid decrease in atomic oxygen density, which is mainly due to the vertical transport and chemical losses. At an altitude of around 85 km, atomic oxygen density already declines by one order of magnitude to $10^{10}$ atoms cm$^{-3}$. The typical value of simultaneously retrieved spectral resolution is around 0.48 nm, and no wavelength shift is found.

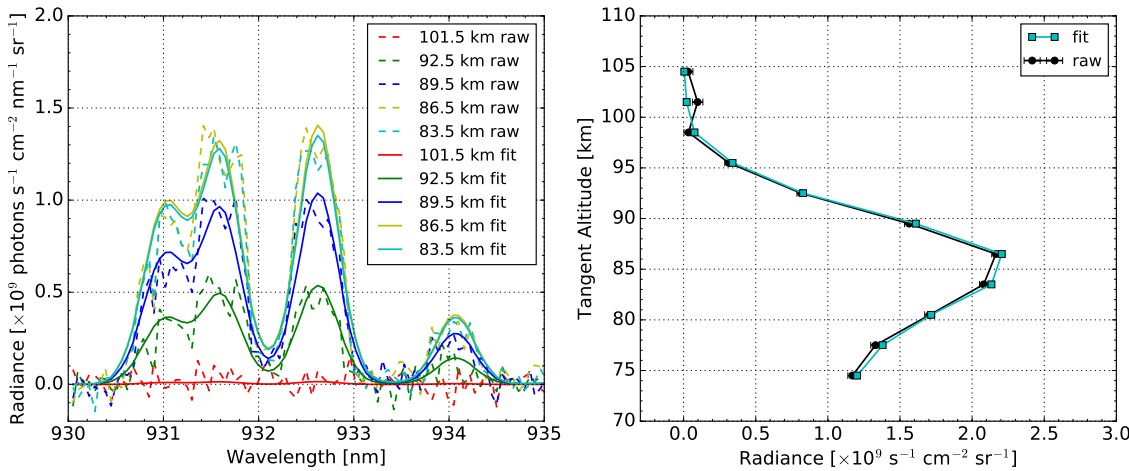

**Figure 5.** (Left) Simulated spectra (fit, solid line) and measurements (raw, dashed line) of GOMOS monthly zonal mean measurements of OH(8–4) airglow emissions at tangent altitudes, as given in the figure legend for Aug. 2003 at 30°–40°S and a local time of 10–12 p.m.. (Right) The spectrally integrated radiance over 930–935 nm versus tangent altitude for the same conditions.

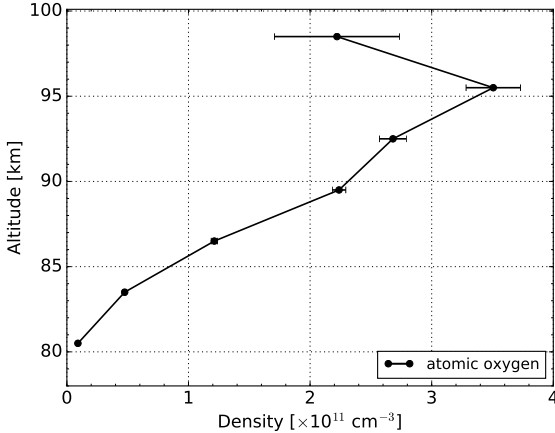

**Figure 6.** Atomic oxygen abundances, derived from GOMOS monthly zonal mean measurements of OH(8–4) airglow emissions for Feb. 2006 at 10°–20°N. The error bar represents the statistical uncertainty coming from the measurement noise. It increases towards higher altitudes, as a consequence of the corresponding SNR being lesser.

## 4.2 Error Analysis

The total uncertainty of the derived atomic oxygen densities not only depends on the measurement noise, but also on the smoothing error as well as on uncertainties in forward model parameters and the background atmosphere input. The largest

source of uncertainties is found in the forward model parameters. The influence of these uncertainties on the results are assessed through error propagation, by the perturbation of forward model parameters. The chemical reaction rate coefficient $k_1$ has an uncertainty of around 20%, contributing around 15% uncertainty below 90 km and around 20% at 95 km in derived abundances. $k_3$ introduces an increasing uncertainty of up to 6% at 95 km. The nascent branching factor (e.g. $f_8$, $f_9$) explains the distribution ratio of excited hydroxyl radicals $OH^*$ of different vibrational levels. $f_8$ has a linear influence on the uncertainty of the results; a perturbation of 10% on its values results in a similar retrieval uncertainty. The errors of Einstein coefficients correspond to an uncertainty of around 7% in the results. The uncertainty in the quenching coefficient $k_{N_2(8)}$ of $OH^*$ radicals with nitrogen molecules introduces a uncertainty of 14% at 80 km, which decreases to 5% at 95 km, and the uncertainty in the rate coefficient for quenching by molecular oxygen $k_{O_2(8)}$ corresponds to an uncertainty of 5% at 85 km and 2% at 95 km. The influences of other model parameters are on the order of 1–2% or less. SABER temperature uncertainties are the predominant factor influencing the retrieval results in the background atmosphere. The uncertainties are around 5.5 K at 80 km and increase to 13 K at 90 km (Dawkins et al., 2018). Through error propagation calculation, this could lead to an uncertainty of 5% below 90 km and up to 20% above 95 km, taking into account the compensation effects of total density changes following the hydrostatic equilibrium (Zhu and Kaufmann, 2018).

At the altitude of 80–100 km, the effects of the smoothing error and measurement noise on the uncertainty are on the order of around 0.5% and 5%, respectively. It is due to a properly chosen regularization in the retrieval procedure that the a priori information is negligible in the retrieval results. As part of a more in-depth look into the retrieval results, the averaging kernel and vertical resolution are investigated, as shown in Figure 7. The summed-up averaging kernels for each row in the altitude region of interest (80–100 km) are equal to one, indicating that the measurements instead of the a priori information contribute to nearly all of the retrieval result. The peaks of averaging kernels are found at the tangent altitudes and the corresponding vertical resolution for each altitude is around 3 km, which is close to the vertical spacing of the limb measurements. Since the sum of the averaging kernels is also near one, the a priori influence is generally low.

## 4.3 Spatial and Temporal Analysis

Atomic oxygen reveals a two-cell structure near 95 km at mid-latitudes, which is most pronounced during the equinox seasons (Figure 8). The smallest values appear over the equatorial region and the largest values are at mid-latitudes. As already mentioned and discussed by Smith et al. (2010) and Xu et al. (2010), the latitudinal distribution structure of atomic oxygen is influenced by tides. The vertical transport of air caused by tides leads to a vertical displacement of atomic oxygen. At a local time of almost midnight (the mean local time of the GOMOS measurements is around 11 p.m., 10 p.m. to 12 p.m.), the atomic oxygen displacement by tides at the mesopause is upward at the equator (resulting in an [O] decrease) and downward in subtropical latitudes (resulting in an [O] enrichment).

In Figure 9, a vertical distribution comparison of derived densities from 2002 to 2011 over the mid-latitude and equatorial regions is shown. Both the annual oscillation (AO) and semiannual oscillation (SAO) can be seen from the temporal evolution of mid and lower latitudes. The SAO reaches its maximum at equinox seasons, which is related to the semiannual variation of the atmospheric tide amplitudes (Smith et al., 2010).

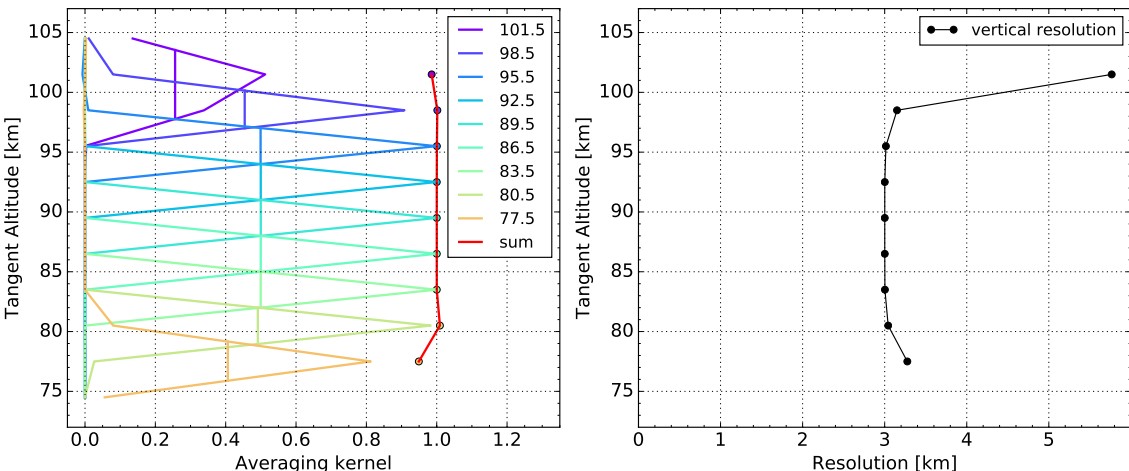

**Figure 7.** (Left) The averaging kernel and (right) the vertical resolution of the retrieval for Feb. 2006 at $10°$–$20°$N and a local time of 10–12 p.m.. The vertical resolution is obtained from the distribution of each row in the averaging kernel by calculating the corresponding FWHM.

A multiple linear regression analysis is applied to quantitatively analyze the longtime variations of the GOMOS [O] dataset. The monthly mean column density integrated from 80–97 km for $20°$–$30°$N is analyzed by harmonic fitting, which includes components such as the solar cycle effect, SAO, AO and QBO (quasi biennial oscillation), and baseline.

$$[O]_{Column} = baseline + A_{solar} \cdot I_{solar}(t + shift) + A_{SAO} \cdot cos(\frac{2\pi t}{6} + P_{SAO})$$

$$+ A_{AO} \cdot cos(\frac{2\pi t}{12} + P_{AO}) + A_{QBO} \cdot cos(\frac{2\pi t}{27.5} + P_{QBO}) \tag{1}$$

The variable t represents the month since 2002 and the column density is fitted by amplitudes (A, atoms cm$^{-3}$) and phase shifts (P, months) of SAO (period of 6 months), AO (period of 12 months), and QBO (period of 27.5 months), complemented by the amplitude (A$_{solar}$, atoms cm$^{-3}$ sfu$^{-1}$) and shift of the 11-year solar cycle effect, as well as a baseline. The coefficient $I_{solar}$ is the solar radio flux proxy ($F_{10.7}$ cm, in unit of sfu) taken from Tapping (2013). Typical mesospheric QBO (MQBO) period is about 27.5 months by investigating mesospheric zonal wind measurements (Ratnam et al., 2008; de Wit et al., 2013;
Malhotra et al., 2016). The baseline is given as the averaged value of the monthly mean column densities along the time series. The non-linear least squares fitting method (Levenberg–Marquardt algorithm) is applied to derive these fitting parameters, as described in detail by Kaufmann et al. (2013) and Zhu et al. (2015).

In Figure 10, the raw data and the fitting results are illustrated in the upper panel. Besides, the baseline plus the solar terms are also shown in the plot. The SolarMin and SolarMax values denote the fitted atomic oxygen column densities solely
from the solar cycle component, under the solar minimum and solar maximum conditions, respectively. The SAO and AO components from the harmonic fitting are given in the middle and lower panel respectively. The [O] longtime variations are well characterized by the fit. The 11-year solar cycle effect is captured, in which the atomic oxygen density is 17% smaller in

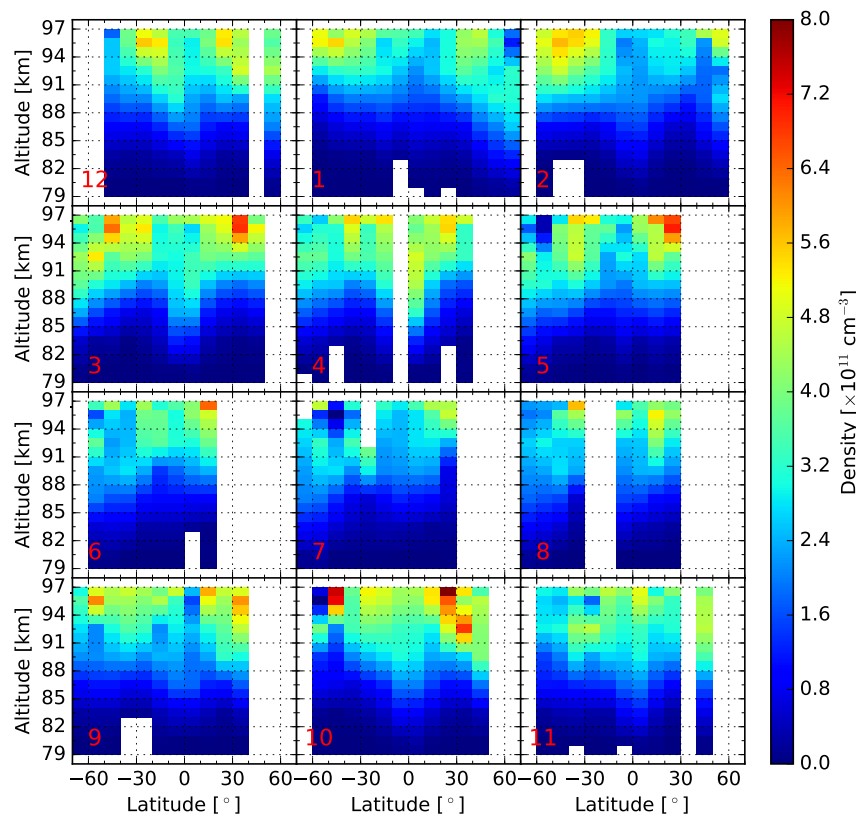

**Figure 8.** Latitude-altitude distribution of the zonal mean atomic oxygen density for 2007. Each row represents approximately a season. The data are linearly interpolated into a 1 km altitude grid for better illustration. The numbers in the subplots indicate the month of the year.

2008/2009 (minimum of solar cycle 23/24) than in 2002 (near solar maximum conditions of solar cycles 23), due to different radiative forcing conditions during the solar cycles. This agrees with model investigations and experimental results, which are normally in a range of around 10% to 30% (Schmidt et al., 2006; Marsh et al., 2007; Kaufmann et al., 2014; Zhu et al., 2015). A significant semiannual oscillation is observed, reaching a maximum in equinox seasons, which is in agreement with the analysis above for Figure 9, and the amplitude is about 18% (with respect to the baseline). The annual oscillation has an amplitude of 10%, with the maximum being reached near summer solstices and the minimum near winter solstices. These results are consistent with the analyses of Zhu et al. (2015) and Lednyts'kyy et al. (2017), which reported SAO amplitudes on the order of 15% and 12%, AO amplitude of 11% and 7%, respectively. The QBO amplitude is on the order of 2%. The multiple linear fitting analyses on other latitudinal bands and altitudes also show a similar solar cycle effect as well as AO and SAO variations, as some examples are summarized in Table 2.

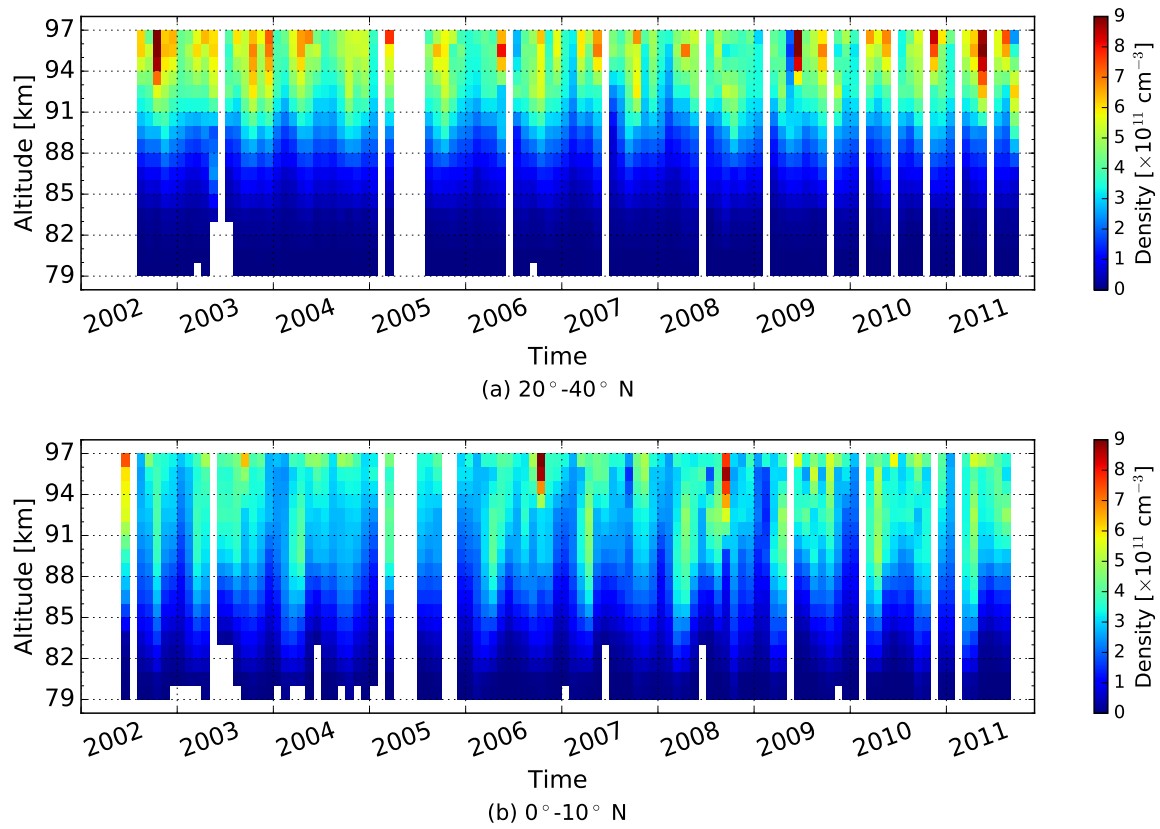

**Figure 9.** Temporal evolution of the vertical distribution of monthly zonal mean atomic oxygen densities for $20°–40°$N (a) and $0°–10°$N (b). The data are linearly interpolated into a 1 km altitude grid.

**Table 2.** Summary of multiple linear regression analysis results of monthly mean atomic oxygen column densities integrated over 80–97 km for $20°–30°$N, $0°–10°$N, and $30°–20°$S from 2002 to 2011. The quantities are in unit of $10^{12}$ atoms $cm^{-3}$. The Solar Min and Solar Max values denote the fitted atomic oxygen column densities solely from the solar cycle component, under the solar minimum and solar maximum conditions respectively. $A_{SAO}$, $A_{AO}$, and $A_{QBO}$ are the amplitudes of SAO, AO, and QBO, respectively.

| Latitude bin | Baseline | Solar Max | Solar Min | $A_{SAO}$ | $A_{AO}$ | $A_{QBO}$ |
|---|---|---|---|---|---|---|
| $20°–30°$N | 1.276 | 0.151 | -0.067 | 0.23 | 0.122 | 0.022 |
| $0°–10°$N | 1.221 | 0.085 | -0.044 | 0.272 | 0.126 | 0.05 |
| $30°–20°$S | 1.126 | 0.097 | -0.052 | 0.183 | 0.074 | 0.025 |

It could be considered to add an additional slope term in the harmonic fitting (Equation 1) as well. In that case, the agreement between measurements and the fit increases marginally by about 2% with an additional slope term. But the fitting parameters are not independent any longer, because a strong correlation between the slope, the baseline and the solar terms are found, which was not the case before. This indicates that the inversion problem (to obtain the fitting parameters) is now under-

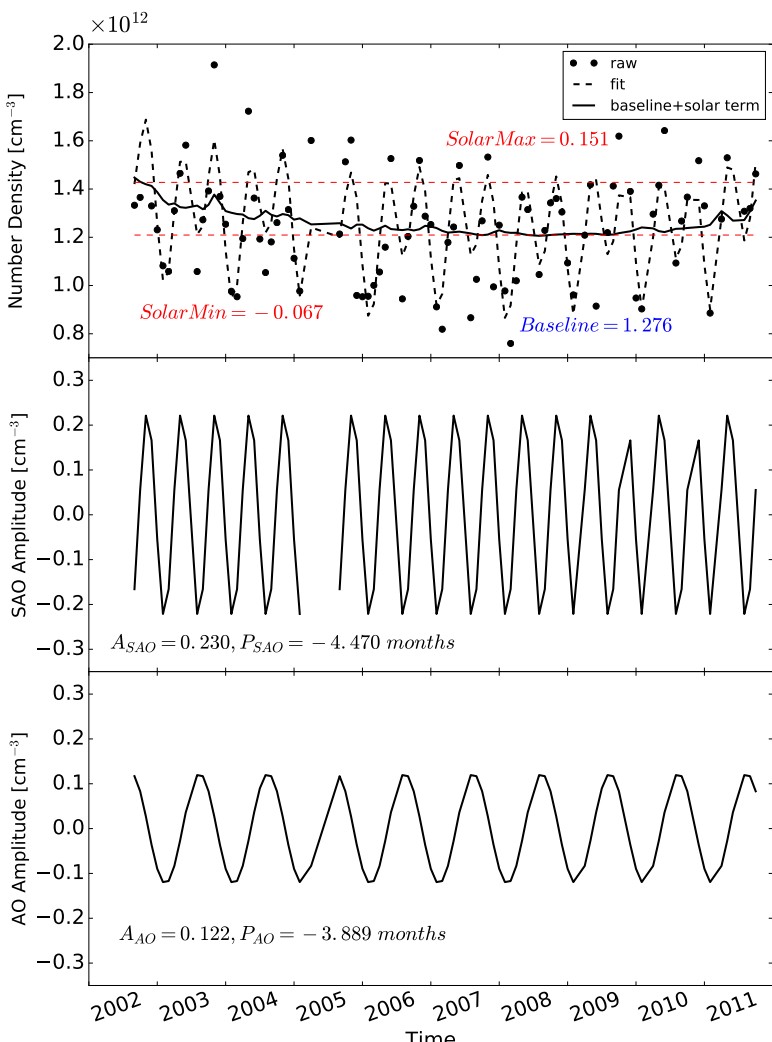

**Figure 10.** Multiple linear regression analysis of vertically integrated, monthly mean atomic oxygen densities of 80–97 km for 20°–30°N from 2002 to 2011. (upper panel) The raw and fitted data are shown along with the baseline plus the solar components in the multiple linear regression results. The SolarMin and SolarMax values denote the fitted atomic oxygen column densities solely from the solar cycle component, under the solar minimum and solar maximum conditions respectively. (middle) SAO and (lower) AO parts are also illustrated with corresponding parameters. $A_{SAO}$, $P_{SAO}$, $A_{AO}$, and $P_{AO}$ are the amplitudes and phase shifts of SAO and AO, respectively. The quantities of SolarMin, SolarMax, baseline and amplitudes are in unit of $10^{12}$ atoms cm$^{-3}$. The phase shifts are in unit of months. The gap present in the SAO is caused by the data discontinuity.

determined. As an alternative approach, the solar ($F_{10.7}$) fitting parameter could be replaced by the slope term. In this case, the residual increases by about 5% and the fitting parameters are not correlated (except for the offset and slope terms). From a mathematical point of view, this is an alternative to the original fit (with solar, but without slope terms). For this setup, the

slope is $-0.0002 \times 10^{12}$ cm$^{-3}$ month$^{-1}$, which means that there is virtually no trend apparent in the data. This can be explained, if the change over time is considered as a combination of two linear trends, with a negative slope in the declining phase of the solar cycle and a positive slope in the following inclining phase. This hypothesis can be underpinned by looking at a subset of the time series, covering the time period from 2002 to 2009, only (roughly solar maximum to solar minimum). The slope for this period is about -3% per year, indicating a linear decrease of atomic oxygen by 21% for the given period. If the $F_{10.7}$ dependency is considered instead, a similar drop is modeled, if a solar term with an amplitude of $0.0025 \times 10^{12}$ cm$^{-3}$ sfu$^{-1}$ is used. This value is similar to $0.002 \times 10^{12}$ cm$^{-3}$ sfu$^{-1}$, which is the value obtained when the total time series is considered. This line of arguments indicates that there is more likely a solar $F_{10.7}$ dependency apparent in the data than a plain linear dependency.

## 5 Discussion

### 5.1 Comparison with SCIAMACHY Data

The SCIAMACHY instrument, another limb sounder on board the Envisat satellite, observed OH emissions at various wavelengths from visible to infrared emissions (Bovensmann et al., 1999; Kaufmann et al., 2008). This provides us with the best opportunity for a comprehensive joint investigation of the GOMOS [O] dataset, as SCIAMACHY covers exactly the same OH(8–4) band wavelengths as GOMOS. Two more datasets of [O] derived from SCIAMACHY green line emissions (Kaufmann et al., 2014; Zhu et al., 2015) and OH(9–6) band airglow (Zhu and Kaufmann, 2018) are currently available.

SCIAMACHY performed the OH airglow measurements in dark limb-viewing mode in the flight direction, with the recorded spectra always near a local solar time of 10 p.m. and a fixed altitude grid of 3.3 km. The OH(8–4) band observation is located in Channel 5 with a spectral resolution of 0.54 nm. SCIAMACHY data Version 8–2016 is adopted in this work. A continuous observation was performed during the entire lifetime of Envisat. The number of recorded profiles in one sample bin was around 100–300 before 2005 and significantly increased to 400–600 because of a change in instrument operations. SNRs of single profiles are normally on the order of 6 at peak altitudes and decrease to 1 at lower altitudes. After monthly zonal averaging, SNRs increase by one order of magnitude, and the mean noise level is around $0.6 \times 10^9$ photons s$^{-1}$ cm$^{-2}$ nm$^{-1}$ sr$^{-1}$.

Theoretically, the SCIAMACHY and GOMOS measurements should be identical in the same wavelength range. In practice, however, due to effects of various factors, such as instrument characteristics, radiometric calibration and fields-of-view, they do not fully conform with each other in terms of absolute radiance or instrument line shapes. In this study, the two data products are found to be consistent in terms of absolute radiance within $\pm$ 20% in the peak emissions layer (shown in Figure 11), after monthly zonal averaging. Particularly, the difference of the GOMOS data to the SCIAMACHY data is gradually becoming positive from negative over time, and the potential source for the drift could be a degradation of the GOMOS or SCIAMACHY instruments (Bramstedt et al., 2009), which is not fully corrected or over-compensated in the level-0 calibration, and the change of the system sensitivities over time. One specific example of spectra comparison is given in Figure 12. The emission radiances from two data products are similar, but the GOMOS spectra are more noisy.

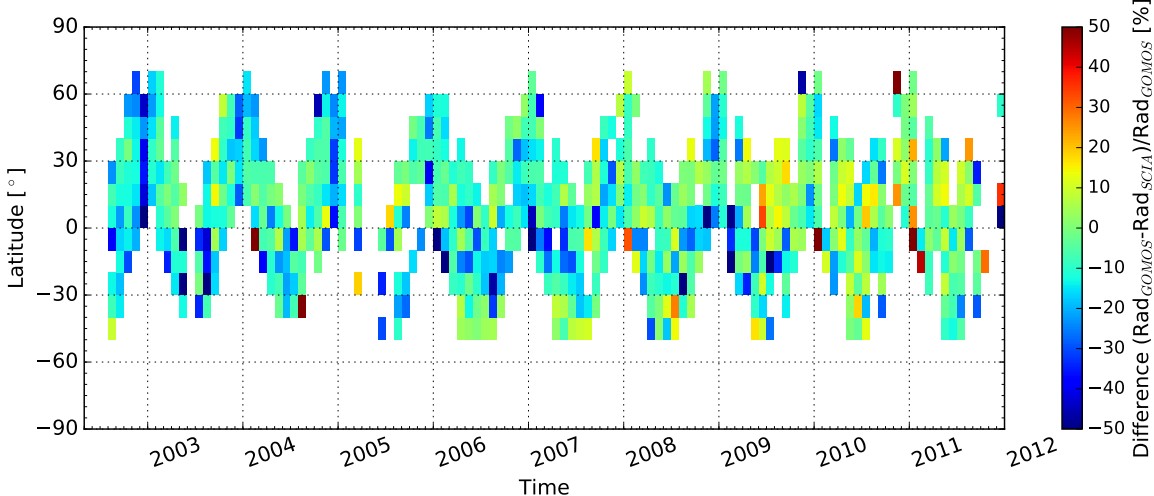

**Figure 11.** Temporal evolution of radiance differences (in percentage) between GOMOS and SCIAMACHY at a tangent altitude of 86.5 km. The radiance is integrated over the wavelength of 930–935 nm. Negative numbers indicate that SCIAMACHY radiances are larger than those of GOMOS.

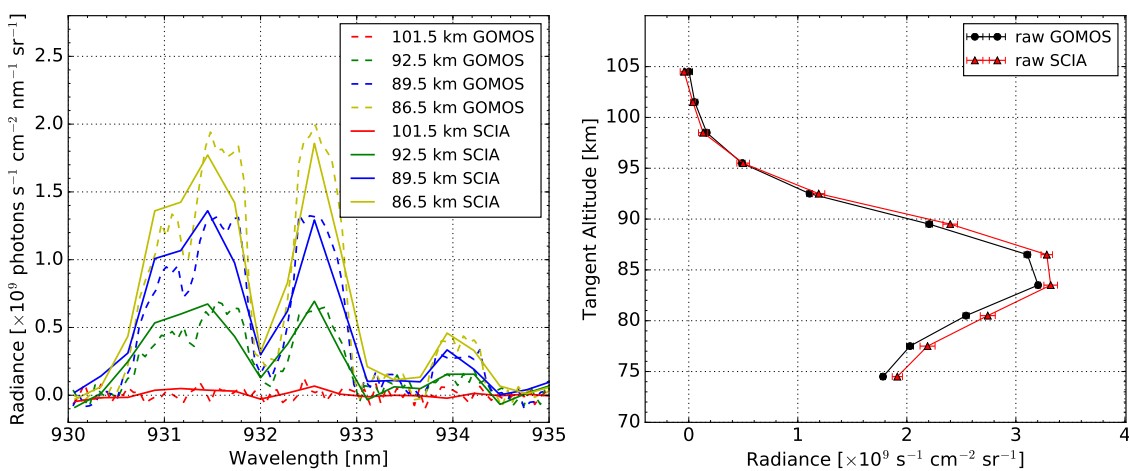

**Figure 12.** (Left) SCIAMACHY (solid line) and GOMOS (dashed line) observations of monthly zonal mean OH(8–4) airglow emissions at the tangent altitudes, as given in the Figure legend for Apr. 2004 at $20°$–$30°$N and a local time of 10–12 p.m. (Right) The spectrally integrated radiance over 930–935 nm versus tangent altitude for the same conditions. The error bars are measurement noise, computed as in Figure 4.

The same retrieval procedure is applied to the SCIAMACHY data. The differences between the atomic oxygen abundances from the two instruments are illustrated in Figure 13. There are no major systematic discrepancies and they agree within a $\pm$ 20% difference in most latitude-altitude bins as expected from the differences of the corresponding radiances. The GOMOS

data is found to be over 20% lower in the northern hemisphere in February, and also in tropical regions in March, May, and September. GOMOS values appear to be 20% larger at low altitudes of around 80 km in some scattered bins. In general, these two atomic oxygen datasets derived from OH(8–4) airglow emissions agree with each other within the combined uncertainties in the context of absolute abundances.

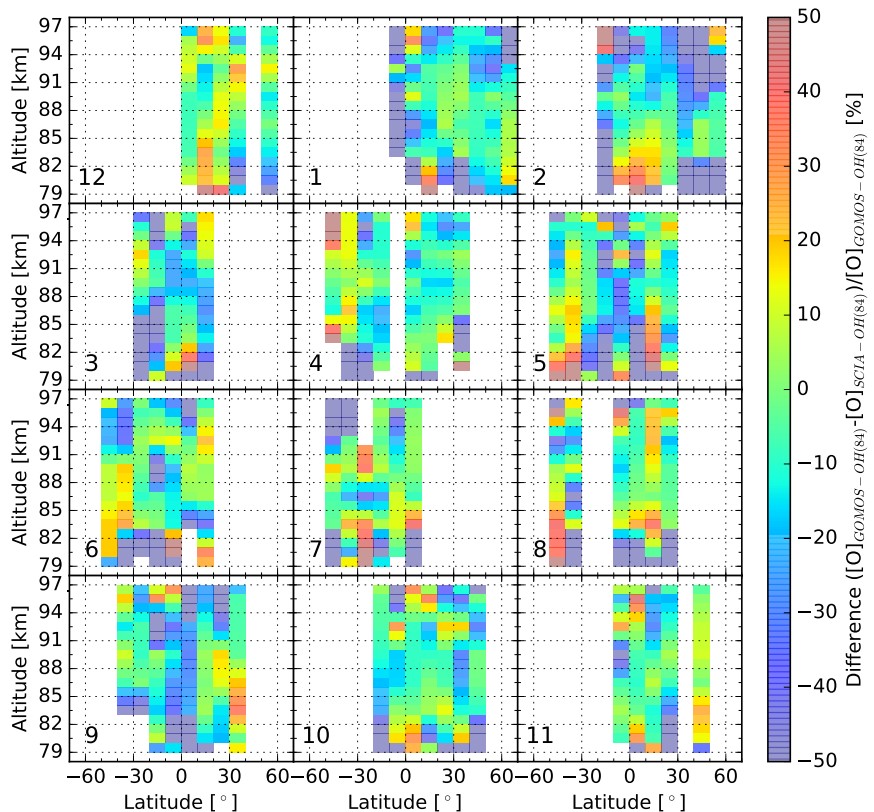

**Figure 13.** Latitude-altitude distribution of percentage differences between zonal mean atomic oxygen densities derived from GOMOS and SCIAMACHY OH(8–4) airglow emissions for 2007. Each row represents approximately a season. Negative numbers indicate that SCIAMACHY abundances are larger than those obtained from GOMOS. The data are linearly interpolated into a 1 km altitude grid. The numbers in the subplots indicate the month of the year.

5    Similarly, a latitude-altitude comparison of the GOMOS data with atomic oxygen obtained from SCIAMACHY OH(9–6) emissions (Zhu and Kaufmann, 2018) is given in Figure 14 for 2007. In general, these two dataset agree with each other, but the GOMOS OH(8-4) dataset is found to be around 10–20% lower than the SCIAMACHY OH(9-6) dataset in most latitude bins, especially in the altitude region of 85-95 km. The difference between the two datasets becomes more than 20% at some

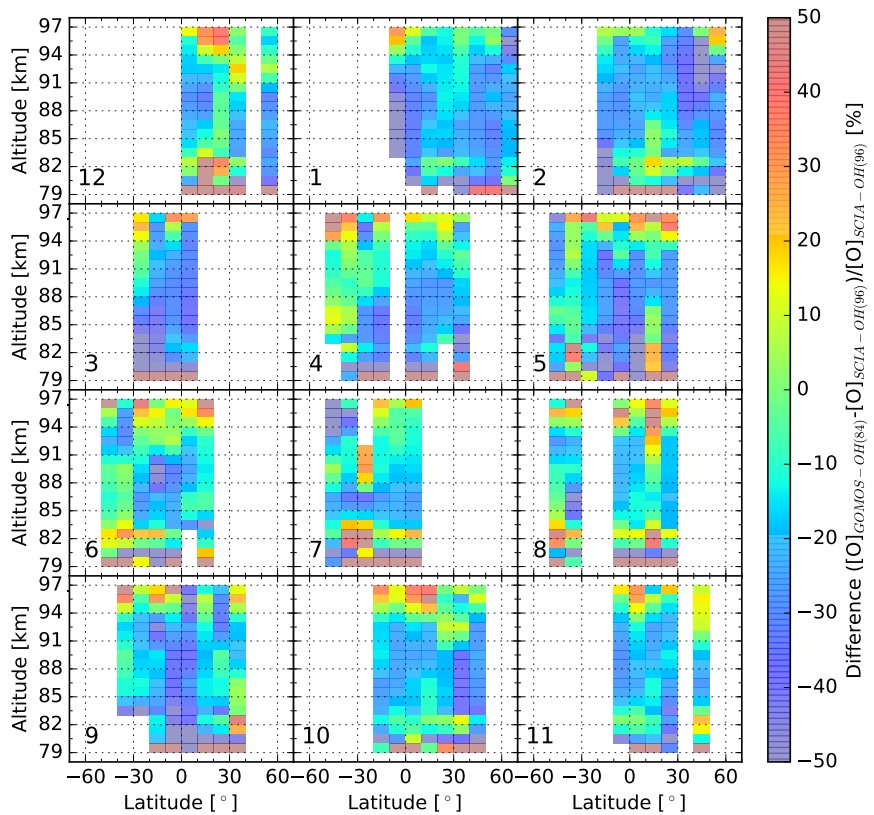

**Figure 14.** Latitude-altitude distribution of percentage differences between zonal mean atomic oxygen densities derived from GOMOS OH(8–4) and SCIAMACHY OH(9–6) airglow emissions for 2007. The SCIAMACHY OH(9–6) are taken from Zhu and Kaufmann (2018). This figure is plotted in a way similar to Figure 13. Negative numbers indicate that SCIAMACHY OH(9–6) atomic oxygen abundances are larger than the GOMOS OH(8–4) abundances.

data points near the equator in March, May and September. Combing the derived results from SCIAMACHY OH(8-4), an inter-comparison of the three datasets is given in Figure 15 for different latitudinal and seasonal conditions. The absolute abundances of the three datasets are in the same order of magnitude and they agree with each other at the altitude region of interest of 80–95 km. Specifically, atomic oxygen abundances derived from OH(8–4) emissions by both instruments are found to be 10–20% lower than those derived from OH(9–6) at around 90 km. This might be explained by a slight underestimation of the quenching of OH(v=9) to OH(v=8) by $O_2$, an overestimation of the deactivation of OH(v=8) due to collisions with atomic or molecular oxygen, or the over/under-estimation of the branching factors $f_9$ and $f_8$ in the OH airglow model.

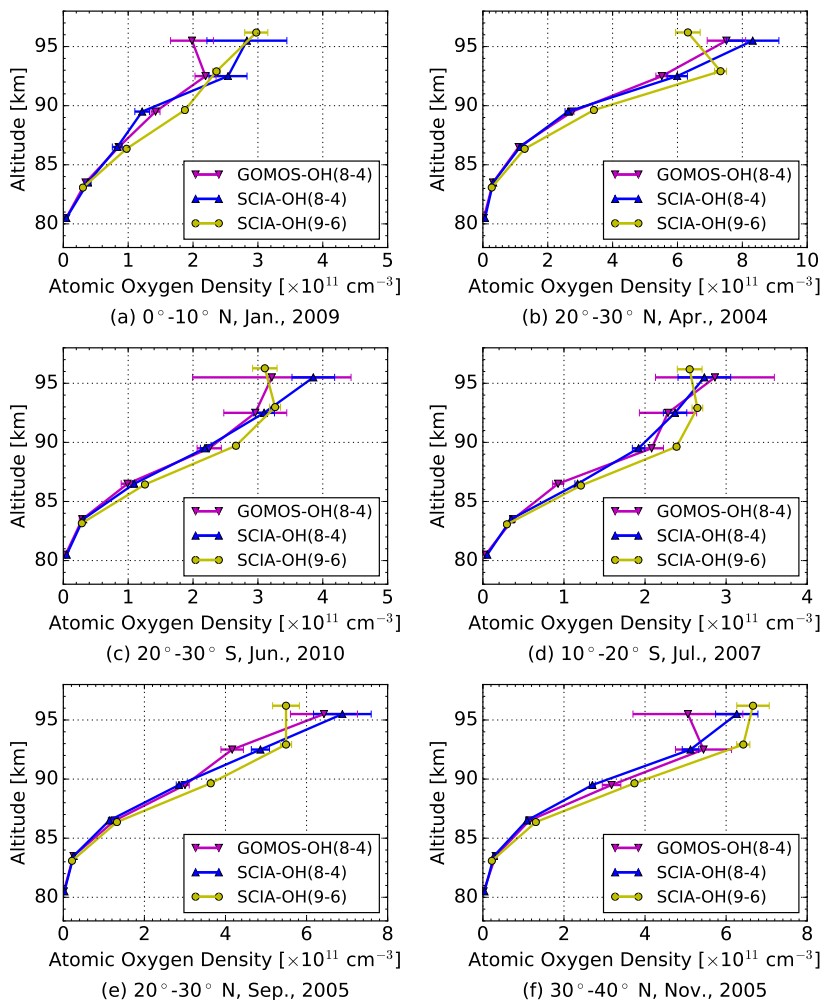

**Figure 15.** Comparison of monthly zonal mean atomic oxygen densities derived from hydroxyl airglow emissions observed by the GOMOS and SCIAMACHY instruments in various latitude bins for different months. SCIA-OH(9–6) represents the atomic oxygen dataset derived from the SCIAMACHY OH(9–6) band by Zhu and Kaufmann (2018); SCIA-OH(8–4) is the dataset from the SCIAMACHY OH(8–4) band; and GOMOS-OH(8–4) is from the GOMOS measurements of the OH(8–4) band.

## 5.2 Comparison with Other Datasets

There are a number of $O_2$ and O excited state emissions, which can also be used to derive atomic oxygen. This includes $O(^1S)$ green line and $O_2$ A-band emissions. Their modeling is mostly independent from the calculation of OH(v) emissions, although some processes have to be considered in all models. Rocket-borne in situ measurements of atomic oxygen are the
5 most independent from methods based on nightglow. Mostly performed in the 1970s (e.g. Dickinson et al., 1980; Sharp, 1980; Offermann et al., 1981), the measurements are very rare and selective in terms of the local time and location. Figure 16 gives

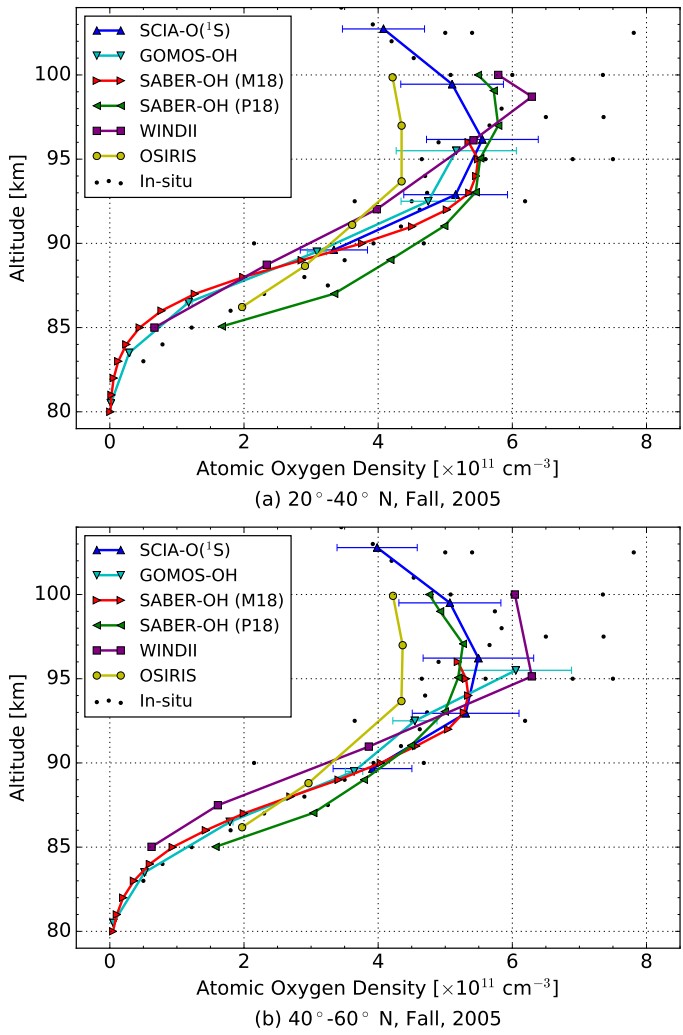

**Figure 16.** Comparison of derived atomic oxygen densities derived from various instruments and measurement techniques averaged for 20°–40°N (a) and 40°–60°N (b) in autumn (Sep., Oct., Nov.), 2005. SCIA-O($^1$S) is the atomic oxygen data derived from SCIAMACHY green line emissions (Kaufmann et al., 2014; Zhu et al., 2015); SABER-OH refers to the atomic oxygen datasets derived by Mlynczak et al. (2018) (M18); and Panka et al. (2018) (P18) from SABER hydroxyl airglow emissions. The WINDII dataset is obtained from WINDII combined hydroxyl and green line observations, 1993 (Russell and Lowe, 2003; Russell et al., 2005), while the OSIRIS dataset is derived from OSIRIS $O_2$ A band measurements (Sheese et al., 2011). In situ data are obtained from rocket-borne experiments with mass spectrometers, conducted at different local times at 37–40°N, from 1972 to 1976 (Offermann and Grossmann, 1973; Trinks et al., 1978; Offermann et al., 1981).

an impression of how the various datasets of atomic oxygen available in the literature fit to each other. All the sets are selected

using a similar local time of around 10–11 p.m., with the exception of OSIRIS (6:30 p.m.) and in situ data with diverse local times at midnight or in the afternoon.

The datasets agree within their combined uncertainties in most cases. The absolute abundances are typically $4–6 \times 10^{11}$ atoms cm$^{-3}$ above 90 km and decrease with descending altitudes by one order of magnitude (at around 80 km) for mid-latitude in autumn. GOMOS data are around 10% lower than SCIAMACHY-O($^1$S) and both SABER-OH datasets at 90–95 km, but remain in good agreement with these datasets at lower altitudes below 90 km. The OSIRIS dataset appears as the lower bound of the values above 90 km, as it is always the lowest in this region, while it becomes relatively large below 90 km. The WINDII dataset is around 10% lower than the GOMOS-OH data at an altitude of 87–92 km, but they generally fit to each other. In situ data scatter in a large variation, which might be caused by the diurnal tides (local time differences), and the GOMOS-OH dataset is still located in its overall range of spread.

## 6 Conclusions

GOMOS limb observations of the background atmosphere provide the opportunity to retrieve atomic oxygen abundances from hydroxyl nightglow emissions at the mesopause. A global night-time [O] dataset is obtained by applying the OH modeling and retrieval method to the monthly zonal mean of GOMOS limb measurements, with the atmospheric background profiles of temperature, total density and ozone taken from the SABER measurements. Its uncertainty comes from the measurement noise (around 5%), selected relaxation schemes and kinetic parameters in OH modeling (contributing around 20% in total) and background atmosphere inputs, for example atmospheric temperature, ozone (around 5% to 20%). The obtained profiles present an overall picture of the vertical distribution of atomic oxygen from 80 km to 100 km. A temporal analysis of the profiles shows 11-year solar cycle effect tendencies as well as semiannual and annual variations, of which SAO is the most prominent.

The GOMOS data agrees with the SCIAMACHY OH(8–4) measurements, with deviations typically smaller than 20%. They are, on average, about 10–20% lower than atomic oxygen data obtained from SCIAMACHY OH(9–6) observations. This might indicate that the collisional energy exchange between OH(v=9) and OH(v=8) via collisions with $O_2$ and O in the OH airglow model requires some readjustments. Compared to other datasets derived from various instrument measurements, such as SABER, WINDII, OSIRIS and in situ rocket experiments, the GOMOS data also demonstrates an agreement with these datasets within their combined uncertainties.

*Data availability.* The GOMOS and SCIAMACHY data used in this study are available to the public at ftp://eoa-dp.eo.esa.int as part of the Cat-1 project 2515. SABER Version 2.0 can be downloaded from http://saber.gats-inc.com. Derived atomic oxygen datasets are available on request.

## Appendix A:  Fitting of the selected collisional rate coefficients for OH(v=8)

Some rate coefficients used in this work are obtained by simultaneously fitting the OH airglow model to measured limb radiances of OH(9-6) and OH(8-5) bands. The measurements are taken from SCIAMACHY channel 6 radiances. The OH(9-6) band radiance is integrated over the wavelength range of 1378-1404 nm, and the OH(8-5) band is integrated over 1297-1326 nm. The selected parameters are adjusted in such a way that the ratio between the simulated radiances of the two bands is consistent with the ratio obtained from the measurements. Several cases with different rate coefficients or combinations being adjusted in the fitting are considered, as given in the Table A1. The fitting results of different cases are illustrated in the Figure A1, as compared to the SCIAMACHY measurements. Model simulations of cases b, c and d give a good agreement with the measurements. The rate coefficient for the collisional removal of OH(v=8) by atomic oxygen differs by nearly one order of magnitude from literature (Xu et al., 2012), and the fitted parameters should agree with the laboratory measurements within the combined uncertainties if available (Dyer et al., 1997). Therefore the case b is chosen and applied in this work. The utilization of cases c and d will influence the retrieval results, that atomic oxygen abundances will differ by around 5% above 90 km and 15% at 80 km compared to case b.

**Table A1.** The comparison of study cases with the applied rate coefficients being summarized. The adjusted parameters and their fitted values are marked bold, while the coefficients taken from laboratory measurements are underlined and marked italic.

| Study case | $k_{O(8)} \times 10^{-10} cm^3 s^{-1}$ | $k_{O_2(8)} \times 10^{-12} cm^3 s^{-1}$ | $k_{N_2(8)} \times 10^{-13} cm^3 s^{-1}$ | $k_{O_2(9,8)} \times 10^{-13} cm^3 s^{-1}$ |
|---|---|---|---|---|
| a | 1.2 | _8.0_ | _7.0_ | 42.0 |
| b | **0.65** | _8.0_ | _7.0_ | **8.9** |
| c | **0.25** | **12.0** | _7.0_ | 42.0 |
| d | **0.35** | _8.0_ | **15.0** | 42.0 |
| e | 2.0 | **5.5** | _7.0_ | 42.0 |
| f | 2.0 | _8.0_ | **4.5** | 42.0 |

*Author contributions.*  QC processed the data, performed the analysis and drafted the manuscript. MK and YZ initiated the topic, provided insight and instructions, and discussed the results regularly. All authors contributed to the revision and improvement of the paper.

*Competing interests.*  The authors declare that they have no conflict of interest.

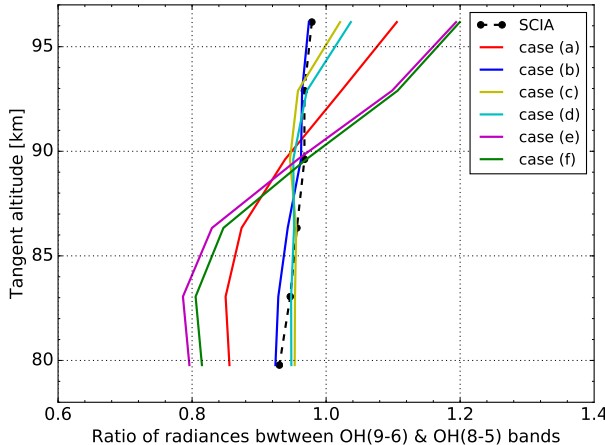

**Figure A1.** The ratio of the integrated limb radiances between the OH(9–6) (1378–1404 nm) and OH(8–5) (1297–1326 nm) bands versus tangent altitude. The raw data (black dashed line) is taken from the SCIAMACHY channel 6 measurements. The fitted results (solid line) are obtained by applying the rate coefficients with respect to different cases.

*Acknowledgements.* Q. Chen was supported in her work by the China Scholarship Council. The work of Y. Zhu was supported by the 2017 Helmholtz–OCPC Programme and the International Postdoctoral Exchange Fellowship Program 2017. We also thank three anonymous referees for their valuable comments and suggestions.

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
