# Peer review of "Global nighttime atomic oxygen abundances from GOMOS hydroxyl airglow measurements in the mesopause region"

_Atmospheric Chemistry and Physics, 2019_

## Referee Comment (RC1) · Anonymous Referee #1 · 18 May 2019

Review on the manuscript "Global nighttime atomic oxygen abundances from resampled GOMOS hydroxyl airglow measurements in the mesopause region" by Chen et al.

The manuscript reported a new dataset of nighttime time atomic oxygen density in the mesopause region derived from the GOMOS instrument on board Envisat. The general features of atomic oxygen can be found, such the peak height and the peak number density etc. The errors, which are introduced by various sources (e.g., chemical reaction rate coefficients, Einstein coefficients, quenching coefficient, atmospheric temperature and other parameters), are analyzed quantively. The spatial and tempo-

ral variations are presented and consistent with previous studies. Finally, the validation is performed by comparing with other independent measurements. A general consistency is obtained among various datasets.

This study provided a new data set of atomic oxygen in the mesopause region. It is very useful in the atmospheric study community.

The paper is well written and may become acceptable for publication. But there are some minor comments noted below, which should to be addressed/incorporated.

CommentsïijŽ

1. Cross comparisons with other data sets are very important. This reviewer found that the O profile of 20-30N, Apr., 2004 in Figure 13 is very consistent with the results of Xu et al., [2012] (see Figure 5). Figures 5 and 6 in Xu et al., [2012] show that the retrieved O profiles at north hemisphere, south hemisphere and equator are very different, which dependent on seasons. How about GOMOS measurements? The comparison and discussions should be added.

Xu, J., H. Gao, A. K. Smith, and Y. Zhu (2012), Using TIMED/SABER nightglow observations to investigate hydroxyl emission mechanisms in the mesopause region, J. Geophys. Res., 117, D02301, doi:10.1029/2011JD016342.

2. P2 L1-2: "although it is difficult to obtain a consistent global picture of absolute density values from these measurements, which differ by a factor of more than 40"ïijŇ What are values of the absolute density from these measurements? How about the local time and locations of these measurements? 3. P2 L35: "GOMOS"→"GOMOS (Global Ozone Monitoring by Occultation of Stars) ". 4. P9: Do these errors relate to latitudes? 5. P10 L4-6: "migrating diurnal tides, which have a maximum wind amplitude over the equator and two weaker maxima of opposite signs at mid-latitudes." Which component of wind? Zonal or meridional wind or vertical wind? How about the peak height of the migrating tides in wind? This might be the fundamental for the downward

and/or upward displacement of [O]. Please clarify the procedure. 6. P10: Equation (1), What is the physical meaning of "Offset"? 7. P12 L6-7: "18%" and "9.6%", How about the baseline of this percentage?
* * *

---

## Referee Comment (RC2) · Anonymous Referee #2 · 11 Jun 2019

This paper describes how GOMOS observations of radiances in the OH(8-4) band are used to derive a new [O] product, a very important species in the mesosphere-lower thermosphere. It is presented in straightforward and easy to follow manner and is fairly well written. I believe it would be a suitable topic for ACP and would recommend it for publication after some mostly minor details (discussed below) are properly addressed.

General point: In my opinion, the introductory material focuses too much on the level 1 data and not nearly enough on the retrieval (level 1 to level 2) process. It might not be necessary to cut text about the level 1 data, but much more detail about the retrieval algorithm itself is needed, e.g. what kind of retrieval algorithm is it (as Rogers outlines
many different schemes)? What did you use for your a priori? What assumptions are made in the forward model that could lead to potential uncertainties in the l2 data?

Specific points:

P1 Line 3: "based on" should be "derived from".

Line 22: there should be "e.g." at the start of the reference list.

P3 Line 2: OH(v=8) should be OH(8-4)

Line 12: please change "entities" to some thing like trace species, or atmospheric states. Also, what is meant by "mostly"?

Line 14: "descending node equator crossing time" is redundant.

Line 23: "lights" should be "emissions"

P4 Line 19: please, briefly, explain "star leakage" in the text.

P5 Line 12: what is meant by "quantified"?

P6 Fig 3: "altitude" should be either "tangent altitude" or "tangent height".

Fig 4: If I'm understanding this correctly you're showing slant column radiances, in which case it should really be "tangent height" (this would also need to be corrected in the caption and anywhere else it is discussed).

P7 Line 10: there are many approaches given in Rogers. As your technique hasn't been published before, you need to have a lot more detail here about the retrieval algorithm.

Line 11: it is unclear what "complemented by variables" means. As it is, it sounds like the intention is to retrieve, e.g., wavelength shift parameters. Are you retrieving those? Or are you just trying to say that there is a wavelength shift taken into account in the spectral fitting?

Line 15: where is your a priori coming from? If you say that it's coming from the "real atmosphere" that implies that you already know what the true state is. Your a priori needs to be described in much more detail.

P8 Line 3: "longtime" should be "time"

Fig 5 (right): I get somewhat nervous when I see measured and fitted profiles that agree that well. How did you choose to show this specific profile? Is this a typical fit, or is it one of the best fits? Some discussion needs to go in to why this example was chosen. Also, the y-axis is labelled as "altitude". Again, if I'm understanding this correctly you're showing slant column radiances, in which case it should really be "tangent height" (also needs to be corrected in the caption).

Line 15: "accuracy" is probably not the correct term here (implies systematic error). I think you're discussing "total uncertainty".

P9 Fig 6 caption: "lower" should be "lesser"

Lines 1-10: this section just needs a few lines of text describing how the error estimates were calculated (presumably through error propagation).

Multiple instances: "average kernel" should be "averaging kernel"

Lines 17-18: it is not the fact that the vertical resolution is close to the retrieval grid that tells you that the results is coming from the measurements and not the a priori. That information comes from the sum of the averaging kernels (which you have plotted and is near 1). Please discuss this.

P10 Fig 7 caption: "The retrieval altitude grid refers to the vertical resolution of the derived quantities". It's unclear what you were trying to say here.

Eq 1: It may not be strictly necessary, but why are you only fitting to an offset and not an offset + slope? I assume the slope wouldn't be significant, but it would be nice to have for completeness.

P11 Lines 9-10: Please rephrase this sentence, as it doesn't fully or accurately define what is being shown in Fig 10.

P12 Fig 9 caption: could be good to remind people that the time bins are monthly means

Line 1: I could easily be wrong, but I doubt that the uncertainty on the solar cycle fit is precise to 0.1%. If it is, fine. But if not, I'd recommend rounding 17.1% to 17% (same with 9.6% at line 7).

P13 Fig 10: the AO and SAO panels are missing units (relative amplitude, I assume), same for the given P values (months, I assume).

Fig 10 and Table 2: It is unclear what the values for Solar Max and Solar Min represent. Please explain in the text and captions.

P14 Line 1: This may be a matter of opinion, but "validation" requires in depth analysis of comparisons with multiple other instruments. I would recommend changing the name of this section to "comparisons with SCIAMACHY".

Line 9: Did SCIA "always" measure at 10PM? Surely, that must change somewhat with latitude? Perhaps "always measured near 10pm"?

Line 17: could perhaps also mention fields-of-view.

Line 27: Please clarify what you mean by "consistent". Do you mean agree within the combined uncertainties?

Lines 32-33 (and on next page): When you use "OH(v=8) emissions", it makes me think that you're measuring multiple OH(v=8) bands, when I believe you're only looking at 8-4 (same with the v=9 instances). Please use the more specific notation. Also, at some point early on you should define that OH(8-4) means OH(v'=8 -> v"=4).

P15 Fig 11: One thing that immediately grabs your attention in this plot is that the difference is becoming noticeably more positive over time, which seems to indicate

that there's a drift in one, or both, of the data sets. Can you please comment in the text what might be the source of this change over time?

P16 Line 2: I don't think I agree with "as proxies for"; I would agree with "to derive". And to keep this sentence consistent with the next, I'd suggest instead of "excited states" you could have "excited state emissions".

Line 5: There are many more references available. At the very least you need an "e.g." before the reference.

Line 6: what is meant by "selective"?

P19 In the conclusions, please also summarize how the new data set compares with those from other instruments.
* * *

---

## Referee Comment (RC3) · Anonymous Referee #3 · 15 Jun 2019

The paper presents a new dataset of atomic oxygen measurements in the MLT region. Atomic oxygen is a key species for understanding this atmospheric region and its current measurements all show, probably associated to its variability, very large uncertainties. Therefore, a new measurement database is very important.

I have found the paper very clearly written, with an extensive background, and the methodology and measurements description very well covered and written. It is very comprehensive since it covers from the retrieval to its analysis and validation, including a comparison with several atomic oxygen datasets.

I therefore recommend the paper for publication in ACP. I am listing, though, some

comments below which the authors might want to consider for improving the paper.

Probably my major comment is one that I have already given in a similar paper. The authors named the retrieved atomic oxygen as "GOMOS-OH" O, when it is actually additionally based on the external measurements of three key atmospheric quantities: O3, temperature and pressure (density). The authors clearly state this in the introduction and also present an error analysis of these parameters in the retrieved O but, in my opinion, it should also be mentioned in the abstract and in the conclusion. In this way the reader would have a more clear idea of the derived data.

Other comments, in order of appearance in the manuscript (not in order of importance) are listed below.

Title: Is really need the word "resampled"?

Page 1, l. 13. For the sake of clarity I would write " ... by the photolysis of molecular oxygen and of ozone..."

Page 5. Interval of wavelength used. The authors justify the limited spectral interval because of problems at longer wavelengths. They state errors of about 12% at these wavelengths. However, from the spectrum shown in Fig. 3, which looks very good at wavelengths of 935-955 nm, and given that 12% is not really large compared to the overall error of about 20%, I wonder if it would not have been useful to include the wider spectral interval. If not, could this be considered as an additional complexity (potential source of error) of retrieving O from the OH Meinel bands?

Page 6. Figure caption. Last two lines. Could the authors clarify (give some more details or even an equation) about how the measurement noise of the mean spectra (those used as measurements in the retrieval) were computed? In particular, the mention "integrating", over which quantity? "residual noise standard deviation" residual of what? "spectrum"? I believe they where averaging the spectra in the month/latitude bins, correct? I have no doubt the authors are doing correctly but it would be useful a

more detailed description for the readers. Also, please consider moving it to the body text.

Page 7, line 3. "total"? Do they want to say something additional to "the removal of OH(v=8) by O"?

Also, I do not understand why adjusting ONLY this rate for "adjusting the OH(v=8) populations to be consistent with laboratory measurements? Why not, for example, adjusting other rates as, e.g., that with O2? I might be wrong but this seems to me like as ad hoc adjustment with not much justification. Are these uncertainties included in the model error budget?

How this rate (OH(v=8) +O) compares to that used/derived by Sharma et al and Panka et al.? In general, could the authors comment on the similarities/differences of the rates affecting OH(8) with those used in other O retrievals from OH? In the conclusion section they mention the possibilities of the discrepancies of this O dataset with other databases possibly caused by differences in the collisional rates. I have no doubt of that but the discussion suggested would be very beneficial to support such conclusion.

The measurements from Oliva et al. are not laboratory measurements but ground-based nadir atmospheric observations, aren't they?

Page 7, line 15. "... the a priori information from the real atmospheric state.."? Please clarify: the "real" atmospheric O is used as a priori? This does not make sense. Please be more specific, which O is used as a priori? (I understand from the following text that a priori has a negligible effect but this should be clarified).

lines 16-17. Which kind (order) of Tikhonov regularization? "Noise is minimized...", which is then the vertical resolution of the retrieved O? This is clearly discussed below, but if you talk about the "noise" at this point you should also mention the vertical resolution.

Page 10, lines 6-7. At which altitude does the discussion in the paragraph above these

lines refer to? At the O peak near 95 km? Note that the opposite behaviour is observed (larger values near the equator) in Fig. 8 for, e.g., moths 3, 4, 10 and 11, at an altitude near 90 km. The reader might be confused, as I was.

Which is the mean local time for each of the plots? Does it change with latitude?

Page 12, line 3. "... solar cycles", caption of Fig. 10, Table 2 and the whole discussion of the solar cycle. The study about the solar cycle(s) has been done assuming that the latest GOMOS measurements analysed, December of 2011, coincides with he maximum of solar cycle 24. However, this is not fully correct, see https://www.swpc.noaa.gov/products/solar-cycle-progression. This shows that the maximum can be placed somewhere between the 2nd half of 2012 or more likely near the end of 2014. Hence, I suggest that all the discussion, including the two suggested (soalrmax and soalrmin amplitudes) be revised accordingly. The data presented do not really cover a full solar cycle (see https://www.swpc.noaa.gov/news/solar-cycle-24-status-and-solar-cycle-25-upcoming-forecast).

Page 14, line 7. Although mentioned earlier it is very useful to give here the references to the O databases.

Lines 20-21. I would remove the last sentence: "The O ... shown in Fig. 13b". First, because it is not discussed here but later, and also because it additionally contains another O-retrieved profile which is discussed in a different paragraph.

I would exchange the order of Figs. 13 and 14.

Fig. 13 is very important as it compares the three O- retrievals. However, for the community, it is more interesting to provide a figure of a GLOBAL comparison of the already published SCIAMACHY OH(9-6) O dataset by Zhu and Kaufmann (2018) with he current dataset. E.g. similar to Fig. 14 but for SCIAMACHY OH(9–6) band instead of SCIAMACHY OH(8-4).

Page 14, line 29. Typo SCIAMACHY

---

## Author Comment (AC1) · 17 Sep 2019

**Answers to the referee comments**

We thank all referees for the valuable and constructive comments. We have addressed all the points one-by-one as listed below. The page/line numbers indicating the changes in the replies are given with respect to the old manuscript, and may differ from the modified one. The manuscript is revised accordingly, with all changes marked using the latexdiff program, and is included in this document after the replies.

**Referee #1**

**1. Cross comparisons with other data sets are very important. This reviewer found that the O profile of 20-30N, Apr., 2004 in Figure 13 is very consistent with the results of Xu et al., [2012] (see Figure 5). Figures 5 and 6 in Xu et al., [2012] show that the retrieved O profiles at north hemisphere, south hemisphere and equator are very different, which dependent on seasons. How about GOMOS measurements? The comparison and discussions should be added.**

According to the referee's comment, we added the following comparison similar to Xu et al. (2012) (see Fig. 5&6). The plot below provides the comparison of derived GOMOS atomic oxygen profiles, at 30°-50°N, 10°S-10°N and 30°-50°S for (a) Mar. and (b) Jul. 2003, respectively.

[Figure]

Figure. The Comparison of derived GOMOS atomic oxygen profiles, at 30° −50° Jul. 2003, respectively. Since the data for 30 ◦ −50 ◦ N, Jul. 2003 is not available in GOMOS, it is substituted by the data from 20° −30° N of this month.

This figure provides the comparison among the derived GOMOS atomic oxygen profiles from the equator, the northern and southern hemispheres, for both the spring equinox (Mar.) and summer solstice (Jul.) seasons. In March, the atomic oxygen abundances at the equator are larger than those at both hemispheres below 92 km, but above they become smaller than the northern and southern hemisphere values. However, in July, the atomic oxygen density is found to be largest in the northern hemisphere above 91 km. The data from Xu et al. (2012, see Fig. 5&6) also shows a similar behaviour, despite of the larger absolute values, which may be due to the larger radiance as measured by SABER.

**2. P2 L1-2: "although it is difficult to obtain a consistent global picture of absolute density values from these measurements, which differ by a factor of more than 40" What are values of the**

**absolute density from these measurements? How about the local time and locations of these measurements?**

Offermann et al., [1981] (see Tab.6 and Fig.7) reviewed the rocket experiments prior to 1980 and summarized the corresponding absolute atomic oxygen density measurements. These measurements were obtained by three kinds of measurement technology, which provide data differing by one order of magnitude. The highest values are delivered by the mass spectrometers with a cryo-ion source, which are $5.8 \sim 9.8 \times 10^{11} cm^{-3}$ at peak height, and those measurements were taken at 37~40°N latitude,at local time of 23:31, 01:25, 20:16 and 15:30. The second group is given by the technique of ultraviolet absorption and resonant scattering, the absolute value of which are about $7.8 \sim 28 \times 10^{11} cm^{-3}$ at peak height. The location of the experiments is at 57°N latitude, and the local time scatters around noon (11:04, 12:56, 13:59) and midnight (22:37, 23:09, 23:55). The last group are provided by the cryo-pumped mass spectrometer, and their values vary around $0.87 \sim 9.1 \times 10^{11} cm^{-3}$ at peak height. They are obtained at 30~38°N latitude, with local time in the afternoon (14:30, 14:32) and near midnight (22:23, 22:33, 22:34, 02:16).

All the measurements were measured in the mid-latitudes of the north hemisphere, and the local time is either near the noon or at the midnight. However the obtained atomic oxygen values diverge by more than one order of magnitude, and even for the same measuring technique, the values are variable. Therefore, Sharp, [1991] concluded it to be "differ over a factor of 40".

We added "Offermann et al., [1981]" also in the reference list for this sentence.

**3. P2 L35: "GOMOS"→"GOMOS (Global Ozone Monitoring by Occultation of Stars)".**

We modified the text according to the referee's comment.

**4. P9: Do these errors relate to latitudes?**

Generally, we do not see a significant latitude dependence of the errors of the derived atomic oxygen from GOMOS measurements.

**5. P10 L4-6: "migrating diurnal tides, which have a maximum wind amplitude over the equator and two weaker maxima of opposite signs at mid-latitudes." Which component of wind? Zonal or meridional wind or vertical wind? How about the peak height of the migrating tides in wind? This might be the fundamental for the downward and/or upward displacement of [O]. Please clarify the procedure.**

As the local time of the GOMOS observation is limited to midnight, it is difficult to deduce migrating diurnal tides from the GOMOS atomic oxygen dataset. Besides, the nonmigrating diurnal tides could also influence the spatial distribution of atomic oxygen and OH emissions (Xu et al. 2010). Therefore, it is difficult to characterize migrating tides as provided by GOMOS.

According to the referee's comment, to clarify the procedure, we modified the sentences in P10, Line 4 as follows:

"As already mentioned and discussed by Smith et al. (2010) and Xu et al. (2010), the latitudinal distribution structure of atomic oxygen is influenced by tides. The vertical transport of air caused by tides leads to a vertical displacement of atomic oxygen."

**6. P10: Equation (1), What is the physical meaning of "Offset"?**

According to the referee's comment, we modified "offset" to "baseline", and added the following sentence in P11 Line 6 to describe its physical meaning:

"The baseline is given as the averaged value of the monthly mean column densities along the time series."

**7. P12 L6-7: "18%" and "9.6%", How about the baseline of this percentage?**

All the percentages, 17.1%, 18%, 9.6% and 1.7% are calculated with respect to the baseline value.

According to the referee's comment, we modified the text in P12 Line 6: "the amplitude is about 18% (with respect to the baseline)"

**Referee #2**

General point:

**In my opinion, the introductory material focuses too much on the level 1 data and not nearly enough on the retrieval (level 1 to level 2) process. It might not be necessary to cut text about the level 1 data, but much more detail about the retrieval algorithm itself is needed, e.g. what kind of retrieval algorithm is it (as Rogers outlines many different schemes)? What did you use for your a priori?**

Our retrieval approach is based on the Gauss-Newton iteration scheme, with the zero- and first- order Tikhonov regularization matrix applied. The a priori data about the absolute value of atomic oxygen is taken from MSIS model, and is averaged into the vertical grid of 3 km as the measurements for the zero-order matrix, and the first-order derivative matrix for smoothing is constructed from the linear gradient of two consecutive atmospheric layers. We modified the paragraph in P7, Line 10-17 to clarify it:

"The inverse model applies a constrained global-fit approach following the formalism of Rodgers (2000). The Gauss-Newton iterative method in the n-form (Rodgers, 2000, p. 85) is chosen to minimize the cost function of this inverse problem. Besides, a priori information about the atmospheric state is included in the retrieval for regularization to mitigate the influence of measurement errors. The a prior information about atomic oxygen in this work is taken from MSIS model data, and the zero- and first- order Tikhonov regularization matrices (Tikhonov and Arsenin, 1977) are considered in the cost function. The a priori data about the absolute value of atomic oxygen is taken from MSIS model, which is averaged into the vertical grid of 3 km as the measurements. The first order regularization is obtained from the linear interpolation of the a priori data given on the measurement grid, i.e., no sub measurement-grid information is obtained from that data. The regularization strength depends on altitude and its main purpose is to assure meaningful values at the upper and lower boundaries of the altitude regime considered. In between, the regularization has virtually no effect on the retrieved quantities, as can be seen from the retrieval diagnostics. The vertical resolution of the retrieval results are close to the vertical grid of the measurements. The target parameters of the retrieval are the vertical profiles of atomic oxygen abundance, spectral resolution, and a wavelength shift. The latter are both altitude-independent and give a better agreement between measured and simulated spectra. The content of information in the spectra is sufficient to retrieve these additional parameters."

**What assumptions are made in the forward model that could lead to potential uncertainties in the l2 data?**

The assumptions in the forward model that could lead to potential uncertainties in the L2 data, are mainly forward model parameters (the reaction and quenching rate coefficients, Einstein coefficients) and the atmospheric background input profiles (temperature, ozone, density).

Specific points:

**1. P1 Line 3: "based on" should be "derived from".**

We modified the text according to the referee's comment.

**2. Line 22: there should be "e.g." at the start of the reference list.**

We added "e.g." in the text according to the referee's comment.

**3. P3 Line 2: OH(v=8) should be OH(8-4).**

We modified the text according to the referee's comment.

**4. Line 12: please change "entities" to some thing like trace species, or atmospheric states.**

We modified "entities" to "trace species and temperature profiles" according to the referee's comment.

**Also, what is meant by "mostly"?**

Besides the stellar transmission measurements, the GOMOS instrument is equipped with two photometers, which deliver the stellar flux radiance measurements. Since the word "mostly" here is unclear, it is deleted in the text.

**5. Line 14: "descending node equator crossing time" is redundant.**

We changed "descending node equator crossing time" to "equator crossing time (descending node)" according to the referee's comment.

**6. Line 23: "lights" should be "emissions".**

We modified the text according to the referee's comment.

**7. P4 Line 19: please, briefly, explain "star leakage" in the text.**

We added the following sentences in the text according to the referee's comment.

"The star is a point source, and part of the stellar light is spread to the lower and upper band, which is supposed to be totally imaged in the central band in an ideal case."

**8. P5 Line 12: what is meant by "quantified"?**

We rephrased the sentence to "the quality of the reprocessed spectra are evaluated" in the text to make it clear.

**9. P6 Fig 3: "altitude" should be either "tangent altitude" or "tangent height".**

We changed "altitude" to "tangent altitude" in the caption according to the referee's comment.

**10. Fig 4: If I'm understanding this correctly you're showing slant column radiances, in which case it should really be "tangent height" (this would also need to be corrected in the caption and anywhere else it is discussed).**

According to the referee's comment, we changed:

The caption and the label of the right plot in Fig.4: "altitude" to "tangent altitude".

P5 Line 16: "at around 85 km" to "at the tangent altitude of around 85 km".

The caption and the label of the right plot in Fig.5: "altitude" to "tangent altitude".

The caption in Fig.11: "altitude" to "tangent altitude".

The caption and the label of the right plot in Fig.12: "altitude" to "tangent altitude".

**11. P7 Line 10: there are many approaches given in Rogers. As your technique hasn't been published before, you need to have a lot more detail here about the retrieval algorithm.**

According to the referee's comment, we added the following sentence in P7 Line 10:

"The Gauss-Newton iterative method in the n-form (Rodgers, 2000, p. 85) is chosen to minimize the cost function of this inverse problem."

**12. Line 11: it is unclear what "complemented by variables" means. As it is, it sounds like the intention is to retrieve, e.g., wavelength shift parameters. Are you retrieving those? Or are you just trying to say that there is a wavelength shift taken into account in the spectral fitting?**

The variables for the spectral resolution and wavelength shift are also simultaneously derived as the atomic oxygen abundances.

According to the referee's comment, we modified the sentence in P7 Line 11 for clarity as:

"The target parameters of the retrieval are the vertical profiles of atomic oxygen abundance, spectral resolution, and a wavelength shift. The latter are both altitude-independent."

**13. Line 15: where is your a priori coming from? If you say that it's coming from the "real atmosphere" that implies that you already know what the true state is. Your a priori needs to be described in much more detail.**

According to the referee's comment, we modified the last paragraph of P7:

"Besides, a priori information about the atmospheric state is included in the retrieval for regularization to mitigate the influence of measurement errors. The a prior information about atomic oxygen in this work is taken from MSIS model data, and the zero- and first- order Tikhonov regularization matrices (Tikhonov and Arsenin, 1977) are considered in the cost function. The a priori data about the absolute value of atomic oxygen is taken from MSIS model, which is averaged into the vertical grid of 3 km as the measurements. The first order regularization is obtained from the linear interpolation of the a priori data given on the measurement grid, i.e., no sub measurement-grid information is obtained from that data. The regularization strength depends on altitude and its main purpose is to assure meaningful values at the upper and lower boundaries of the altitude regime considered. In between, the regularization has virtually no effect on the retrieved quantities, as can be seen from the retrieval diagnostics. "

**14. P8 Line 3: "longtime" should be "time"**

We modified the text according to the referee's comment.

**15. Fig 5 (right): I get somewhat nervous when I see measured and fitted profiles that agree that well. How did you choose to show this specific profile? Is this a typical fit, or is it one of the best fits? Some discussion needs to go in to why this example was chosen.**

The shown plot is considered to be one of the best fit results, that the total radiances of the raw and the fit are nearly identical. We chose to show the plot here with the intention to illustrate what a best fitting case is like.

According to the referee's comment, we changed this sentence in P8, Line 4, and provided a plot showing a typical fit: "Shown in Figure 5 (left) is a typical profile of the fitted spectra compared with the measurements. In general, simulations and measurements are in good agreement. The spectrally integrated radiances in Figure 5 (right) also show consistency."

[Figure]

Figure 5. (Left) Simulated spectra (fit, solid line) and measurements (raw, dashed line) of GOMOS monthly zonal mean measurements of OH(8–4) airglow emissions at tangent altitudes, as given in the figure legend for Aug. 2003 at $30°$–$40°$ S and a local time of 10–12 p.m.. (Right) The spectrally integrated radiance over 930–935 nm versus tangent altitude for the same conditions.

**Also, the y-axis is labelled as "altitude". Again, if I'm understanding this correctly you're showing slant column radiances, in which case it should really be "tangent height" (also needs to be corrected in the caption).**

We changed "altitude" to "tangent altitude" in the plot and the caption according to the referee's comment.

**16. Line 15: "accuracy" is probably not the correct term here (implies systematic error). I think you're discussing "total uncertainty".**

We changed "accuracy" to "total uncertainty" according to the referee's comment.

**17. P9 Fig 6 caption: "lower" should be "lesser".**

We modified the text according to the referee's comment.

**18. Lines 1-10: this section just needs a few lines of text describing how the error estimates were calculated (presumably through error propagation).**

According to the referee's comment, to describe the error estimation process, we added the following sentences in P8 Line 17:

"The influence of these uncertainties on the results are assessed through error propagation, by the perturbation of forward model parameters."

And in P9 Line 8 we added in the sentence: "Through error propagation calculation".

**19. Multiple instances: "average kernel" should be "averaging kernel".**

According to the referee's comment, we changed "average kernel" to "averaging kernel" in the caption of Fig.7, P9 Line 13, Line 14 and Line 16.

**20. Lines 17-18: it is not the fact that the vertical resolution is close to the retrieval grid that tells you that the results is coming from the measurements and not the a priori. That information comes from the sum of the averaging kernels (which you have plotted and is near 1). Please discuss this.**

According to the referee's comment, we modified the sentences in lines16-18 to explain the vertical resolution of the retrieval:

"and the corresponding vertical resolution for each altitude is around 3 km, which is close to the vertical spacing of the limb measurements. Since the sum of the averaging kernels is also near one, the a priori influence is generally low."

**21. P10 Fig 7 caption: "The retrieval altitude grid refers to the vertical resolution of the derived quantities". It's unclear what you were trying to say here.**

According to the referee's comment, for clarity we changed the label in the right plot of Fig.7 to "vertical resolution" and also rephrased the caption:

"The vertical resolution is obtained from the distribution of each row in the averaging kernel by calculating the corresponding FWHM."

**22. Eq 1: It may not be strictly necessary, but why are you only fitting to an offset and not an offset + slope? I assume the slope wouldn't be significant, but it would be nice to have for completeness.**

According to the referee's comment, a slope term is added in the linear fit component for the harmonic analysis, and the fitting result is shown below in the figure and discussed below:

"The agreement between measurements and the fit increases marginally by about 2% with an additional slope term. But the fitting parameters are not independent any longer, because a strong correlation between the slope, the baseline and the solar terms are found (see Figure below), which was not the case before. This indicates that the inversion problem (to obtain the fitting parameters) is now under-determined.

As an alternative approach, the solar (F10.7) fitting parameter could be replaced by the slope term. In this case, the residual increases by about 5% and the fitting parameters are not correlated (except for the offset and slope terms). From a mathematical point of view, this is an alternative to the original fit (with solar, but without slope terms). For this setup, the slope is -0.0002 $\times 10^{12}$ cm$^{-3}$ month$^{-1}$, which means that there is virtually no trend apparent in the data.

This can be explained, if the change over time is considered as a combination of two linear trends, with a negative slope in the declining phase of the solar cycle and a positive slope in the following inclining phase. This hypothesis can be underpinned by looking at a subset of the time series, covering the time period from 2002 to 2009, only (roughly solar maximum to solar minimum). The slope for this period is about -3% per year, indicating a linear decrease of atomic oxygen by 21% for the given period. If the F10.7 dependency is considered instead, a similar drop is modeled, if a solar term with an amplitude of 0.0025 $\times 10^{12}$ cm$^{-3}$ sfu$^{-1}$ is used.

This value is similar to $0.002 \times 10^{12}$ cm$^{-3}$ sfu$^{-1}$, which is the value obtained when the total time series is considered.

To the authors' opinion, this line of arguments indicates that there is more likely a solar F10.7 dependency apparent in the data than a plain linear dependency."

[Figure]

Figure. Multiple linear regression analysis of vertically integrated, monthly mean atomic oxygen densities of 80–97 km for $20°-30°$ N from 2002 to 2011. (first panel) The raw and fitted data are shown. The fitting is obtained by considering the baseline, the solar term, SAO, AO and QBO terms in the harmonic analysis. The baseline plus the solar term is given additionally. (second) A slope term is included in the fitting along with the solar term. (third) The solar term is replaced by a slope.

[Figure]

Figure. The correlation matrix (left) with and (right) without the slope term for full data series

According to the referee's comment, we added the above description at the end of section 4.3.

**23. P11 Lines 9-10: Please rephrase this sentence, as it doesn't fully or accurately define what is being shown in Fig 10.**

According to the referee's comment, we rephrased this sentence to describe the Fig.10:

"In Figure 10, the raw data and the fitting results are illustrated in the upper panel. Besides, the baseline plus the solar terms are also shown in the plot. The SolarMin and SolarMax values denote the fitted atomic oxygen column densities solely from the solar cycle component, under the solar minimum and solar maximum conditions respectively. The SAO and AO components from the harmonic fitting are given in the middle and lower panel respectively."

**24. P12 Fig 9 caption: could be good to remind people that the time bins are monthly means.**

We changed the caption to "Temporal evolution of the vertical distribution of monthly zonal mean atomic oxygen densities" according to the referee's comment.

**25. Line 1: I could easily be wrong, but I doubt that the uncertainty on the solar cycle fit is precise to 0.1%. If it is, fine. But if not, I'd recommend rounding 17.1% to 17% (same with 9.6% at line 7).**

According to the referee's comment, the precision of the fitting results was reconsidered. We rounded up the numbers for the solar cycle, AO and QBO fit components.

P12 Line 1 & Line 3: 17.1% to 17%.

P12 Line 7: 9.6% to 10%.

P12 Line 9: 1.7% to 2%.

**26. P13 Fig 10: the AO and SAO panels are missing units (relative amplitude, I assume), same for the given P values (months, I assume).**

According to the referee's comment, we modified Fig. 10 to include the units of SAO and AO panels, which is the amplitude, and also the P values. We also added in the caption: "The phase shifts are in unit of months."

**27. Fig 10 and Table 2: It is unclear what the values for Solar Max and Solar Min represent. Please explain in the text and captions.**

According to the referee's comment, we added in the text and the captions the explanations about the SolarMin and SolarMax:

"The SolarMin and SolarMax values denote the fitted atomic oxygen column densities solely from the solar cycle component, under the solar minimum and solar maximum conditions respectively."

**28. P14 Line 1: This may be a matter of opinion, but "validation" requires in depth analysis of comparisons with multiple other instruments. I would recommend changing the name of this section to "comparisons with SCIAMACHY".**

We changed "validation by" to "comparison with" according to the referee's comment.

**29. Line 9: Did SCIA "always" measure at 10PM? Surely, that must change somewhat with latitude? Perhaps "always measured near 10pm"?**

The SCIAMACHY instrument do not "always" measure at 10 p.m. but around 10 p.m.. SCIAMACHY recorded a swath of 960 km in horizontal direction and covered local times mostly between 21:00 and 23:00 at tangent points. Besides, the local time varies gradually to an earlier/later time with increasing latitude. We modified the sentence to "always measured near 10 p.m." to make it accurate.

**30. Line 17: could perhaps also mention fields-of-view.**

We added "fields-of-view" into this sentence according to the referee's comment.

**31. Line 27: Please clarify what you mean by "consistent". Do you mean agree within the combined uncertainties?**

We changed the sentence to "agree with each other within the combined uncertainties" in the text to make it clear.

**32. Lines 32–33 (and on next page): When you use "OH(v=8) emissions", it makes me think that you're measuring multiple OH(v=8) bands, when I believe you're only looking at 8-4 (same with the v=9 instances). Please use the more specific notation. Also, at some point early on you should define that OH(8-4) means OH(v'=8 -> v''=4).**

According to the referee's comment, we changed the notation in these sentences from "OH(v=8)" to "OH(8-4)" and "OH(v=9)" to "OH(9-6)". Besides, we added the definition of OH(8-4) in P5 Line 10: "It includes a number of emission lines from OH(v=8-4) band, which originates from the radiative transitions of OH(v'=8–v''=4)."

**33. P15 Fig 11: One thing that immediately grabs your attention in this plot is that the difference is becoming noticeably more positive over time, which seems to indicate that there's a drift in one, or both, of the data sets. Can you please comment in the text what might be the source of this change over time?**

According to the referee's comment, we added in the text (P14 Line 19) to explain the potential source of this change: "Particularly, the difference of the GOMOS data to the SCIAMACHY data is gradually becoming positive from negative over time, and the potential source for the drift could be a degradation of the GOMOS or SCIAMACHY instruments (Bramstedt et al., 2009), which is not fully corrected or over-compensated in the level-0 calibration, and the change of the system sensitivities over time."

**34. P16 Line 2: I don't think I agree with "as proxies for"; I would agree with "to derive". And to keep this sentence consistent with the next, I'd suggest instead of "excited states" you could have "excited state emissions".**

We changed the sentence to "There are a number of $O_2$ and O excited state emissions, which can also be used to derive atomic oxygen" in the text according to the referee's comment.

**35. Line 5: There are many more references available. At the very least you need an "e.g." before the reference.**

According to the referee's comment, we added more references and also "e.g." in this sentence as "Mostly performed in the 1970s (e.g. Dickinson et al., 1980; Sharp, 1980; Offermann et al., 1981)"

**36. Line 6: what is meant by "selective"?**

According to the referee's comment, we modified this sentence to "...and selective in terms of the local time and location" to clarify "selective".

**37. P19 In the conclusions, please also summarize how the new data set compares with those from other instruments.**

According to the referee's comment, we added in the P19 Line 13: "Compared to other datasets derived from various instrument measurements, such as SABER, WINDII, OSIRIS and in situ rocket experiments, the GOMOS data also demonstrates an agreement with these datasets within their combined uncertainties."

**Referee #3**

General point:

**Probably my major comment is one that I have already given in a similar paper. The authors named the retrieved atomic oxygen as "GOMOS-OH" O, when it is actually additionally based on the external measurements of three key atmospheric quantities: O3, temperature and pressure (density). The authors clearly state this in the introduction and also present an error analysis of these parameters in the retrieved O but, in my opinion, it should also be mentioned in the abstract and in the conclusion. In this way the reader would have a more clear idea of the derived data.**

According to the referee's comment, we added the description about the used external measurements in the abstract (P1 Line 4)

 "...on board Envisat, with the SABER measurements for the atmospheric background."

And also in the conclusion (P19 Line 4)

"...limb measurements, with the atmospheric background profiles of temperature, total density and ozone taken from the SABER measurements."

Specific points:

**1. Title: Is really need the word "resampled".**

We deleted "resampled" in the title according to the referee's comment.

**2. Page 1, l. 13. For the sake of clarity I would write " ... by the photolysis of molecular oxygen and of ozone...".**

We changed the sentence to "is mainly produced by the photolysis of molecular oxygen and of ozone" in the text to make it clear.

**3. Page 5. Interval of wavelength used. The authors justify the limited spectral interval because of problems at longer wavelengths. They state errors of about 12% at these wavelengths. However, from the spectrum shown in Fig. 3, which looks very good at wavelengths of 935-955 nm, and given that 12% is not really large compared to the overall error of about 20%, I wonder if it would not have been useful to include the wider spectral interval. If not, could this be considered**

**as an additional complexity (potential source of error) of retrieving O from the OH Meinel bands?**

The pixel response non-uniformity (PRNU) variation is reported to be around 12% averagely for the entire SPB (SPB1: 755–774 nm & SPB2: 925–955 nm). Our spectrum range of interest is in SPB2, and we found the PRNU changes along the wavelength in SPB2, and a systematic error more than 20% appears in the longer wavelength range. With the SCIAMACHY measurements as reference (Bovensmann et al., 1999), as shown in Fig.3, the radiances at the wavelength range of 935-955 nm are always 25-30% lower in all data points. When the entire spectrum of 925-955 nm is applied in the O retrieval, the derived O is always 25% lower than the SCIAMACHY data. Till now there is no reasonable explanation for this systematic difference between GOMOS and SCIAMACHY at the wavelength range of 935-955 nm (Erkki Kyrölä, personal communication, 2019). Hence, we excluded this wider wavelength in the retrieval.

[Figure]

Figure 3. Monthly averaged spectrum from GOMOS (black solid line) for Feb. 2004 at $40\,°−50\,°$N and at an altitude of 89.5 km. Strong emission lines from OH(8-4) band are annotated with the branch and rotational quantum numbers. The wavelength range from 930 nm to 935 nm is selected and used in the retrieval. The corresponding SCIAMACHY data (red dashed line) is also given here for comparison.

According to the referee's comment, we updated Fig.3 and added a few sentence in P5 Line 8: "As shown in Fig.3, in the spectral range of our interest (SPB2), we found the GOMOS data shows a good agreement with the SCIAMACHY data at the spectral range of 930–935 nm, whereas the GOMOS radiances at the wavelength range of 935-955 nm are always 25-30% lower compared to the SCIAMACHY measurements, which is not understood (E. Kyrölä, personal communication, 2019)."

**4. Page 6. Figure caption. Last two lines. Could the authors clarify (give some more details or even an equation) about how the measurement noise of the mean spectra (those used as measurements in the retrieval) were computed? In particular, the mention "integrating", over which quantity? "residual noise standard deviation" residual of what? "spectrum"? I believe they where averaging the spectra in the month/latitude bins, correct? I have no doubt the authors are doing correctly but it would be useful a more detailed description for the readers. Also, please consider moving it to the body text.**

The residual noise refers to the noise in the spectrum in between of the emission lines (e.g. 930-930.5 nm & 933-933.5 nm & 934.5 -935 nm). It is called "residual" because in the data processing, the background radiation from above 110 km and the "base" offset are already removed from the spectrum, and the resulted spectrum noise is therefore called "residual

noise". The procedure to calculate the measurement noise is as follows: calculating the standard deviation of the residual noise outside of the emission lines ($\sigma$) $\rightarrow$ $\sigma$ is assumed for all wavelengths (the measurement noise is stochastic) $\rightarrow$ integrating the radiances over all wavelength points (N) of the spectral region $\rightarrow$ the measurement noise for the integrated radiance is calculated as $\sigma \times \sqrt{N}$ (square root value of the number of points N).

We have updated the error bars for Figure 4, 5 & 12 accordingly.

According to the referee's comment, we moved the description about the measurement noise from the caption of Fig.4 to the body text.

The last sentence in the caption of Fig.4 is modified as: "The error bars indicate the measurement noise for integrated radiance (see text)."

We added in P5 Line 17: "The error bars in the right plot indicate the measurement noise for integrated radiance. The measurement noise is calculated from the standard deviation of the residual noise in the spectral range in between of the emission lines, and assumed to be the same for all wavelengths, as the intensities of remaining weak emission lines from high rotational levels in the spectral region are by several order of magnitude lower and therefore negligible. For the integrated radiance, the measurement noise is increased by a factor of $\sqrt{N}$, and N refers to the number of integrated wavelength points."

**5. Page 7, line 3. "total"? Do they want to say something additional to "the removal of OH(v=8) by O"?**

According to the referee's comment, we modified the sentence as "..., the rate coefficients for the production of OH(v=8) by the collision of OH(v=9) with oxygen, and the collisional removal of OH(v=8) by atomic oxygen..."

**6. Also, I do not understand why adjusting ONLY this rate for "adjusting the OH(v=8) populations to be consistent with laboratory measurements? Why not, for example, adjusting other rates as, e.g., that with O2? I might be wrong but this seems to me like as ad hoc adjustment with not much justification. Are these uncertainties included in the model error budget?**

**How this rate (OH(v=8) +O) compares to that used/derived by Sharma et al and Panka et al.? In general, could the authors comment on the similarities/differences of the rates affecting OH(8) with those used in other O retrievals from OH? In the conclusion section they mention the possibilities of the discrepancies of this O dataset with other databases possibly caused by differences in the collisional rates. I have no doubt of that but the discussion suggested would be very beneficial to support such conclusion.**

(1) Selection of the collisional rate coefficients for OH(v= 8)

A multi-quantum quenching model is applied for the collision of OH radicals with $O_2$, and the single-quantum quenching is used for the collision with $N_2$. The total removal rates of OH(v=9) by $O_2$, $k_{O2(9)}$, $N_2$, $k_{N2(9)}$ and O, $k_{O(9)}$ are taken from the laboratory measurements by Kalogerakis et al. (2011, 2016), following Zhu and Kaufmann (2018). The uncertainty of model parameters used in this work are mainly the rate coefficients of the following processes:

OH(v'=8) + O $\rightarrow$ OH(0$\leq$v''$\leq$3) + O, with corresponding rate coefficient $k_{O(8)}$     (1)

OH(v'=8) + $O_2$ $\rightarrow$ OH(v''<v') + $O_2$, with corresponding rate coefficient $k_{O2(8)}$     (2)

OH(v'=8) + $N_2$ $\rightarrow$ OH(v''=v'-1) + $N_2$, with corresponding rate coefficient $k_{N2(8)}$     (3)

and

OH(v'=9) + $O_2$ $\rightarrow$ OH(v''=8) + $O_2$, with corresponding rate coefficient $k_{O2(9,\ 8)}$     (4)

For the process (1), the determination of the coefficient $k_{O(8)}$ has long been an issue. Some of the reported values for the total removal rate of OH(v=8) by O from the literature are listed below. These values vary from $4.5\times10^{-11}$ cm$^3$ s$^{-1}$ to $2.5\times10^{-10}$ cm$^3$ s$^{-1}$ by nearly one order of magnitude, and currently there are no values being well validated. Therefore, we decided to adjust the rate of this process to bring SCIAMACHY OH(9-6) and OH(8-5) measurements into a consistent picture.

| Rate coefficient $k_{O(8)} \times10^{-10}$ cm$^3$ s$^{-1}$ | Reference | Source | Note |
|---|---|---|---|
| 2.5 | Makhlouf et al. (1995) | Empirical estimation | Other relevant parameters are adjusted |
| 2.0 | Adler-Golden (1997) | Empirical estimation | Other relevant parameters are adjusted |
| 0.45 | Varandas (2004) | Theoretical calculation | |
| 1.0 | Copeland et al. (2006) | Laboratory measurement | |
| 0.5 | Smith et al. (2010) Mlynczak et al. (2013) | Adjustment to SABER measurement | Other relevant parameters are adjusted |
| 0.65 | Xu et al. (2012) | Fitting to SABER measurement | Other relevant parameters are fitted |
| 0.87 $\times(2.3\pm1)$ | Panka et al. (2017, 2018) Kalogerakis (2019) | Adjustment to laboratory measurement | |
| 1.5 | Mlynczak et al. (2018) | Adjustment to SABER measurement | Other relevant parameters are adjusted |

For the process (2), the rate coefficients are available from the theoretical calculation or laboratory measurements. Adler-Golden (1997) derived the values of $k_{O2}$ for every vibrational level as summarized in Table 2, and also provided the laboratory measurements of $k_{O2(8)}$ in Table 1, as measured by Dyer et al. (1997). These two values of $k_{O2(8)}$ do not fit with each other. Xu et al. (2012, Table 3 and Figure 7) applied a fitting parameter of 0.723 on the derived values of by Adler-Golden (1997), and brought them into an agreement with the laboratory measurements. This laboratory value is also applied by Mlynczak et al. (2013, Table 1), therefore we took the value from the laboratory measurements in our model.

| Rate coefficient $k_{O2(8)} \times10^{-12}$ cm$^3$ s$^{-1}$ | Reference | Source |
|---|---|---|
| 8 $\pm1$ | Dyer et al. (1997) (Adler-Golden (1997)) | Laboratory measurement |
| 11.9 | Adler-Golden (1997) | Fitting to airglow measurement |
| 8.6 (11.9$\times$0.723) | Xu et al. (2012) | Fitting to SABER measurement |
| 8.0 | Mlynczak et al. (2013) | Adler-Golden (1997) |

For the process (3), we took the measured value of $k_{N2(8)}$ by Dyer et al. (1997), which is $k_{N2(8)} = 7\pm4\times10^{-13}$ cm$^3$ s$^{-1}$. It is applied by Adler-Golden (1997, Table 1), Xu et al. (2012, Table 3), Smith et al. (2010), and Mlynczak et al. (2013, 2018), and Panka et al. (2017, 2018).

For the process (4), Adler-Golden (1997) provided a value of $41.8\times10^{-13}$ cm$^3$ s$^{-1}$ according to the empirical expression, and Xu et al. (2012) modified it as $30.2\times10^{-13}$ cm$^3$ s$^{-1}$ with a fitting parameter. Smith et al. (2010) and Mlynczak et al. (2013) also applied a value of $42.0\times10^{-13}$ cm$^3$ s$^{-1}$ following Adler-Golden (1997). And we firstly applied the value of $42.0\times10^{-13}$ cm$^3$ s$^{-1}$ in this model.

| Rate coefficient $k_{O2(9,8)} \times10^{-13}$ cm$^3$ s$^{-1}$ | Reference | Source |
|---|---|---|
| 41.8 | Adler-Golden (1997) | Fitting to airglow measurement |
| 30.2 (41.8$\times$0.723) | Xu et al. (2012) | Fitting to SABER measurement |
| 42.0 | Smith et al. (2010) Mlynczak et al. (2013) | Adler-Golden (1997) |

A low temperature scale factor is applied for the rate coefficients of OH quenching with $O_2$ and $N_2$, which is 1.18 and 1.4 (Lacoursiére et al., 2003, Panka et al., 2017), respectively .

(2) Fitting of the selected collisional rate coefficients for OH(v=8)

In this work, specific parameters (e.g. $k_{O(8)}$) are obtained by simultaneously fitting the model to measured limb radiances of OH(9-6) and OH(8-5) bands, as obtained independently from the SCIAMACHY channel 6 measurements. The OH(9-6) band radiance is integrated over the wavelength range of 1378-1404 nm, and the OH(8-5) band is over 1297-1326 nm. The selected parameters are adjusted in such a way that the altitude-resolved ratios of the fitted radiances between the two bands are consistent with the ratios calculated from the measurements. In our study, several cases with different rate coefficients or combinations being adjusted in the fitting are considered.

The comparison of study cases is given in the table below, in which the applied rate coefficients are summarized. The adjusted parameters and their fitted values are marked **bold**, while the coefficients taken from laboratory measurements are underlined and marked *italic*. The fitting results of different cases are illustrated in the figure below, as compared to the SCIAMACHY measurements.

| Study case | $k_{O(8)} \times 10^{-10}$ cm$^3$ s$^{-1}$ | $k_{O2(8)} \times 10^{-12}$ cm$^3$ s$^{-1}$ | $k_{N2(8)} \times 10^{-13}$ cm$^3$ s$^{-1}$ | $k_{O2(9,\,8)} \times 10^{-13}$ cm$^3$ s$^{-1}$ |
|---|---|---|---|---|
| (a) | **1.2** | *8.0* | *7.0* | 42.0 |
| (b) | **0.65** | *8.0* | *7.0* | **8.9** |
| (c) | **0.25** | **12.0** | *7.0* | 42.0 |
| (d) | **0.35** | *8.0* | **15.0** | 42.0 |
| (e) | 2.0 | **5.5** | *7.0* | 42.0 |
| (f) | 2.0 | *8.0* | **4.5** | 42.0 |

[Figure]

Figure. The ratios of the integrated limb radiances between the OH(9-6) (1378-1404 nm) and OH(8-5) (1297-1326 nm) bands versus altitude. The raw data (black dashed line) is taken from the SCIAMACHY channel 6 measurements. The fitted results (solid line) are obtained by applying the rate coefficients with respect to different cases.

By comparison of the six cases deploying different rate coefficients, it is found that:

1. it is not possible to obtain the optimal fitting by only adjusting one parameters with respect to OH(v=8), i.e. case (a), (e), (f);

2. Combination of $k_{O(8)}$ with one any parameter from $k_{O2(9,\,8)}$, $k_{O2(8)}$ and $k_{N2(8)}$, can provide satisfying fitting results, i.e. case (b), (c), (d).

To select the most suitable case of rate coefficients for our work, the following points are considered:

1. It is preferable to apply the fitted value for $k_{O(8)}$, as the values from literature vary by nearly one order of magnitude, and currently there are no values being well validated.

2. The fitted parameters and the laboratory measurements should agree within the combined uncertainties if the laboratory measurements are available. For example, $k_{O2(8)} = 8\pm1\times10^{-12}$ cm$^3$ s$^{-1}$ and $k_{N2(8)} = 7\pm4\times10^{-13}$ cm$^3$ s$^{-1}$, as measured by Dyer et al. (1997).

3. Fitting three parameters simultaneously are not considered in our work.

Therefore, we chose and applied case (b) in our work, which is $k_{O(8)}=0.65\times 10^{-10}$ cm$^3$ s$^{-1}$, and $k_{O2(9, 8)}=8.9\times 10^{-13}$ cm$^3$ s$^{-1}$.

The obtained rate coefficient for process (4) ($k_{O2(9,8)}$) is in the order of $10^{-13}$ cm$^3$ s$^{-1}$, which is nearly two orders of magnitude smaller than the total removal rate of OH(v=9) by O$_2$ ($k_{O2(9)}$) (Kalogerakis et al., 2011), because most of the collisional loss of OH(v=9) appears in multi-quantum steps. Since the production rate of OH(v=9) is assumed to be balanced by its loss, and process (4) leads to an increment of OH(v=8) population, while this increment is less than 1% compared to the OH nascent production of OH(v=8). Therefore, we thought the influence of $k_{O2(9,8)}$ on the population distribution is small that we did not mention this point in the text.

Meanwhile, the process (1), along with process (2) and (3), depopulates the OH(v=8) radicals. In the mesopause region around 90 km, the density of O, O$_2$ and N$_2$ are in the order of $10^{11}$, $10^{13}$ and $10^{13}$ cm$^{-3}$. Depending on the orders of magnitude of the rate coefficients for process (1), $k_{O(8)}$, (2), $k_{O2(8)}$, and (3), $k_{N2(8)}$, the removal of OH(v=8) by O may be comparable to the removal by O$_2$ or N$_2$. In this work, $k_{O(8)}$ is in order of $10^{-11}$ cm$^3$ s$^{-1}$, $k_{O2(8)}$ in $10^{-12}$ cm$^3$ s$^{-1}$ and $k_{O2(8)}$ in $10^{-13}$ cm$^3$ s$^{-1}$. Hence, the influence of the rate coefficient $k_{O(8)}$ on the population distribution of OH(v=8) is significant, when the other model parameters are already justified.

To justify these two coefficients, the sensitivity of the retrieval results to the parameter $k_{O(8)}$ and $k_{O2(9,8)}$ are evaluated in the error analysis, that a 10% perturbation on the parameters lead to a uncertainty of 2%, and 0.5% on the derived O values. Meanwhile, the selection of different fitting coefficients, e.g., the utilization of case c and d will lead to a uncertainty of around 5 % above 90 km and 15 % at 80 km in the derived abundances.

According to the referee's comment, we modified the description about the rate coefficients in the text to make it accurate: "Additionally, the rate coefficients for the production of OH(v=8) by the collision of OH(v=9) with oxygen, and the the collisional removal of OH(v=8) by atomic oxygen are obtained by simultaneously fitting the limb radiances of OH(9-6) and OH(8-5) bands, which are independently taken from the SCIAMACHY measurements. The OH(9-6) band radiance is integrated over the wavelength range of 1378-1404 nm, and the OH(8-5) band is over 1297-1326 nm. These two parameters are adjusted in such a way that the ratio between the fitted radiances of the two bands is consistent with the ratio calculated from the measurements."

We also summarized the above description about the selection of fitting parameters, and added in the Appendix of the text.

A comparison of $k^p_{O(8)}$ (2.0×10$^{-10}$ cm$^3$ s$^{-1}$), $k^m_{O(8)}$ (1.5×10$^{-10}$ cm$^3$ s$^{-1}$) with $k^c_{O(8)}$ (0.65×10$^{-10}$ cm$^3$ s$^{-1}$) is provided in the figure below, as these three values are applied in the OH modeling of this work, and the corresponding ratios of the fitted radiances between the OH(9-6) and (8-5) bands are calculated, respectively. The ratios with both $k^m_{O(8)}$ and $k^p_{O(8)}$ are found to deviate from the measurements. The results indicates there could be a deviation of the population distribution of OH radicals in v=9 and v=8 by applying different rate coefficients, as compared to the SCIAMACHY measurements. Besides, the retrieval results are affected by the selected value of $k_{O(8)}$. When $k^p_{O(8)}$ and $k^m_{O(8)}$ are applied in our work, the derived O results are found up to 35% and 20% at 95 km larger than before.

[Figure]

Figure. The ratio of the integrated limb radiances between the OH(9-6) and OH(8-5) bands versus tangent altitude. The raw data (black dashed line) is taken from the SCIAMACHY channel 6 measurements. The OH(9-6) band limb radiance is integrated over the wavelength range of 1378-1404 nm, and the OH(8-5) band is over 1297-1326 nm. The fitted result (blue solid line) is achieved from the fitting, of which the obtained parameters are applied in this work. The M18 (red solid line) applies the value from Mlynczak et al. (2018) for the rate coefficient of OH(v=8) + O in the OH modeling, and P10 (green solid line) applies the one from Panka et al. (2017, 2018).

**7. The measurements from Oliva et al. are not laboratory measurements but ground-based nadir atmospheric observations, aren't they?**

Yes, they are. According to the referee's comment, we modified the text as in the last answer.

**8. Page 7, line 15. "... the a priori information from the real atmospheric state.."? Please clarify: the "real" atmospheric O is used as a priori? This does not make sense. Please be more specific, which O is used as a priori? (I understand from the following text that a priori has a negligible effect but this should be clarified).**

The a priori information is not from the real atmospheric state, which is unknown in this case. According to the referee's comment, we modified this sentence as:

"The a prior information about atomic oxygen in this work is taken from MSIS model data, and the zero- and first- order Tikhonov regularization matrices (Tikhonov and Arsenin, 1977) are considered in the cost function."

**9. lines 16-17. Which kind (order) of Tikhonov regularization? "Noise is minimized...", which is then the vertical resolution of the retrieved O? This is clearly discussed below, but if you talk about the "noise" at this point you should also mention the vertical resolution.**

According to the referee's comment, we modified the sentence in P7 Line 15 as:

"... the zero- and first- order Tikhonov regularization matrices (Tikhonov and Arsenin, 1977) are considered in the cost function. The a priori data about the absolute value of atomic oxygen is taken from MSIS model, which is averaged into the vertical grid of 3 km as the measurements. The first order regularization is obtained from the linear interpolation of the a priori data given on the measurement grid, i.e., no sub measurement-grid information is obtained from that data."

And we added in Line 17 the description about the vertical resolution:

"The vertical resolution of the retrieval results are close to the vertical grid of the measurements."

**10. Page 10, lines 6-7. At which altitude does the discussion in the paragraph above these lines refer to? At the O peak near 95 km? Note that the opposite behaviour is observed (larger values near the equator) in Fig. 8 for, e.g., moths 3, 4, 10 and 11, at an altitude near 90 km. The reader might be confused, as I was.**

The discussion in this paragraph refers to the altitude range of 92 - 97 km, where the two-cell structure appears.

According to the referee's comment, we added in P10 Line 2: "Atomic oxygen reveals a two-cell structure near 95 km at mid-latitudes..."

And we added in Line 6: "the atomic oxygen displacement by tides at the mesopause is upward..."

**11. Which is the mean local time for each of the plots? Does it change with latitude?**

Generally, the mean local time for each plot is around 11 p.m. (10 p.m. to 12 p.m.). The local time is around 11 p.m. in the tropical region, and gradually changes to an earlier/later time with increasing latitude (e.g. 60°N corresponds to the mean local time of around 10 p.m.). The largest variation of local time is found to be in the polar region, which is not considered in this work. In conclusion, the local time of the plot is near midnight. We added in P10 Line 6: "...a local time of almost midnight (the mean local time of the GOMOS measurements is around 11 p.m., 10 p.m. to 12 p.m.), ..."

**12. Page 12, line 3. "... solar cycles", caption of Fig. 10, Table 2 and the whole discussion of the solar cycle. The study about the solar cycle(s) has been done assuming that the latest GOMOS measurements analysed, December of 2011, coincides with the maximum of solar cycle 24. However, this is not fully correct, see https://www.swpc.noaa.gov/products/solar-cycle-progression. This shows that the maximum can be placed somewhere between the 2nd half of 2012 or more likely near the end of 2014. Hence, I suggest that all the discussion, including the two suggested (solarmax and solarmin amplitudes) be revised accordingly. The data presented do not really cover a full solar cycle (see https://www.swpc.noaa.gov/news/solar-cycle-24-status-and-solar-cycle-25-upcoming-forecast.**

The year 2011 does not coincide with the maximum condition of solar cycle 24, which is more likely to be 2014. Therefore 2011 can not be used as the representative of the solar maximum condition. Since the year 2002 is very close to the maximum condition of solar cycle 23, it could still be used as the proxy. According to the referee's comment, we modified Page 12 Line 2: "...2002 (near solar maximum conditions of solar cycles 23)...".

Actually, the solar maximum condition for the solar cycle analysis in this work was only based on the year 2002, the maximum condition of the solar cycle 23. Therefore, the year 2011 has no influences on the analyzed results of solar maximum and solar minimum amplitudes.

**13. Page 14, line 7. Although mentioned earlier it is very useful to give here the references to the O databases.**

According to the referee's comment, we added the references to the O databases in P14 Line 7: "...derived from SCIAMACHY green line emissions (Kaufmann et al., 2014; Zhu et al., 2015) and OH(9–6) band airglow (Zhu and Kaufmann, 2018) ..."

**14. Lines 20-21. I would remove the last sentence: "The O ... shown in Fig. 13b". First, because it is not discussed here but later, and also because it additionally contains another O-retrieved profile which is discussed in a different paragraph.**

We removed the last sentence in this paragraph according to the referee's comment.

**15. I would exchange the order of Figs. 13 and 14.**

We exchanged the order of Fig.13 and Fig.14 according to the referee's comment.

**16. Fig. 13 is very important as it compares the three O- retrievals. However, for the community, it is more interesting to provide a figure of a GLOBAL comparison of the already published SCIAMACHY OH(9-6) O dataset by Zhu and Kaufmann (2018) with the current dataset. E.g. similar to Fig. 14 but for SCIAMACHY OH(9–6) band instead of SCIAMACHY OH(8-4).**

According to the referee's comment, we added in P14 a global comparison of the SCIAMACHY OH(9-6) by Zhu and Kaufmann (2018) with GOMOS OH(8-4) for 2007 as below:

[Figure]

Figure 14. Latitude-altitude distribution of percentage differences between zonal mean atomic oxygen densities derived from GOMOS OH(8–4) and SCIAMACHY OH(9–6) airglow emissions for 2007. The SCIAMACHY OH(9–6) are taken from Zhu and Kaufmann (2018). This figure is plotted in a way similar to Figure 13. Negative numbers indicate that SCIAMACHY OH(9–6) atomic oxygen abundances are larger than the GOMOS OH(8–4) abundances.

And we added in the text the discussion: "Similarly, a latitude-altitude comparison of the GOMOS data with atomic oxygen obtained from SCIAMACHY OH(9–6) emissions (Zhu and Kaufmann, 2018) is given in Figure 16 for 2007. In general, these two dataset agree with each other, but the GOMOS OH(8-4) dataset is found to be around 10-20% lower than the SCIAMACHY OH(9-6) dataset in most latitude bins, especially in the altitude region of 85-95 km. The difference between the two datasets becomes more than 20% at some data points near the equator in March, May and September."

**17. Page 14, line 29. Typo SCIAMACHY**

We modified the typo according to the referee's comment.

[revised manuscript text omitted]